# Where are we in the search for an Artificial Visual Cortex for Embodied Intelligence?

**Arjun Majumdar**[1]*, **Karmesh Yadav**[1]*, **Sergio Arnaud**[2]*
**Yecheng Jason Ma**[3], **Claire Chen**[4], **Sneha Silwal**[2], **Aryan Jain**[5], **Vincent-Pierre Berges**[2]
**Tingfan Wu**[2], **Jay Vakil**[2], **Pieter Abbeel**[5], **Jitendra Malik**[2,5], **Dhruv Batra**[1,2]
**Yixin Lin**[2]†, **Oleksandr Maksymets**[2]†, **Aravind Rajeswaran**[2]†, **Franziska Meier**[2]†
[1]Georgia Tech, [2]Meta AI, [2]UPenn, [2]Stanford University, [2]UC Berkeley
https://eai-vc.github.io

## Abstract

We present the largest and most comprehensive empirical study of pre-trained visual representations (PVRs) or visual 'foundation models' for Embodied AI. First, we curate CORTEXBENCH, consisting of 17 different tasks spanning locomotion, navigation, dexterous, and mobile manipulation. Next, we systematically evaluate existing PVRs and find that none are universally dominant. To study the effect of pre-training data size and diversity, we combine over $4,000$ hours of egocentric videos from 7 different sources (over $4.3$M images) and ImageNet to train different-sized vision transformers using Masked Auto-Encoding (MAE) on slices of this data. Contrary to inferences from prior work, we find that scaling dataset size and diversity does *not* improve performance universally (but does so on average). Our largest model, named **VC-1**, outperforms all prior PVRs on average but does not universally dominate either. Next, we show that task- or domain-specific adaptation of **VC-1** leads to substantial gains, with **VC-1** (adapted) achieving competitive or superior performance than the best known results on all of the benchmarks in CORTEXBENCH. Finally, we present real-world hardware experiments, in which **VC-1** and **VC-1** (adapted) outperform the strongest pre-existing PVR. Overall, this paper presents no new techniques but a rigorous systematic evaluation, a broad set of findings about PVRs (that in some cases, refute those made in narrow domains in prior work), and open-sourced code and models (that required over 10,000 GPU-hours to train) for the benefit of the research community.

## 1  Introduction

The visual cortex is a region of an organism's brain, which together with the motor cortex, enables sight to be converted into movement. In this work, we ask the same question that Fukushima [1, 2] asked nearly 50 years ago – how do we design an *artificial visual cortex*, the module in a computational system that (together with a policy) enables an agent to convert camera input into actions? In contemporary AI, this question has been operationalized as the design of pre-trained visual representations (PVRs) or visual 'foundation models' for embodied AI (EAI).[3] Indeed, recent work has shown that PVRs can substantially improve performance and learning efficiency for navigation [3–5] and manipulation tasks [6–9]. Unfortunately, prior studies are incommensurable – using different self-supervised learning (SSL) algorithms on different pre-training datasets, designed

---

*Equal Contribution

†Equal Contribution

[3]We use embodied AI (EAI) as an umbrella term for all communities studying visuomotor control such as robot learning, vision-based reinforcement learning, egocentric computer vision, etc.

37th Conference on Neural Information Processing Systems (NeurIPS 2023).

for, and evaluated on different downstream EAI tasks. Naturally, one might ask: *Does an artificial visual cortex already exist?*[4]

To answer this question, we conduct the most comprehensive empirical study of visual foundation models for EAI to-date. We curate CORTEXBENCH, a benchmark for evaluating PVRs, consisting of 17 tasks spanning low-level locomotion [10], table-top manipulation [11], dexterous manipulation [12], multi-finger coordinated manipulation [13], indoor visual navigation [14], and mobile manipulation [15]. The visual environments span from flat infinite planes to table-top settings to photorealistic 3D scans of real-world indoor spaces. The agent embodiments vary from stationary arms to dexterous hands to articulated mobile manipulators. The learning conditions vary from few-shot imitation learning to large-scale reinforcement learning. The exhaustiveness of this study enables us to draw conclusions with unprecedented scope and confidence.

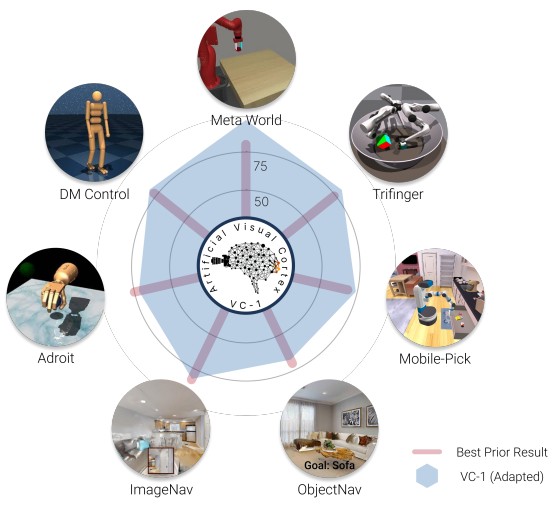

CortexBench

Figure 1: An artificial visual cortex for embodied intelligence must support a diverse range of sensorimotor skills, environments, and embodiments; we curate CORTEXBENCH to systematically measure progress towards this ambitious goal. Our strongest model, denoted **VC-1** (adapted), is ***competitive with or outperforms the best prior results (success rates) on all benchmarks*** in CORTEXBENCH. This comparison is particularly unforgiving because best prior results are benchmark-specific and not constrained to share any aspect of their design.

Our first finding is a *negative result*. We discover that while existing PVRs generally outperform learning-from-scratch baselines, no single PVR is universally dominant. Instead, we find that PVRs tend to work best in the domains (navigation, manipulation etc.) they were originally designed for. We note that no claims of universality were made in prior work, so this finding is illustrative rather than refutative, but nonetheless a significant finding that was not known *a priori*. Overall, serendipity did not come to pass – an artificial visual cortex does not already exist.[2] However, curiously, the *kinds of PVRs* that are locally-dominant in CORTEXBENCH differ significantly in the size and type of pre-training datasets: CLIP [16] used 400M image-text pairs from the web; MVP [8] used 4.5M frames from web-images and many egocentric-video datasets – yet, each performs best on some subset of tasks in CORTEXBENCH. This leads to a natural question: *how does scaling model size, dataset size, or diversity affect performance on* CORTEXBENCH*?* Can we use scaling as a means to learn a single PVR that works for all of the diverse tasks in CORTEXBENCH?

To study these questions, we combine over 4,000 hours of egocentric videos from 7 sources containing humans manipulating objects and navigating indoor spaces, together with ImageNet. From this union, we create 4 pre-training datasets of varying size and diversity, with the largest containing over 5.6M images. We train vision transformers (ViT-B and ViT-L) [17] on these 4 datasets using Masked Auto-Encoding (MAE) [18], and systematically analyze their performance on CORTEXBENCH. To benefit the EAI community, we will open-source these models, which required over 10,000 GPU hours to train.

We do find evidence supporting the scaling hypothesis, but the picture that emerges is more nuanced than what a superficial reading might suggest. Our largest model trained on all data, named **VC-1**, outperforms the best existing PVR by 1.2% on average. However, **VC-1** does *not* universally dominate either – i.e., there are PVRs trained on smaller amounts of data that outperform it on specific tasks. A similar trend emerges for data diversity – more is better on average, but not universally. For instance, the best performance on the `Mobile-Pick` task from Habitat 2.0 [15] is achieved by pre-training on the subset of video data focused on manipulation; presumably because the mobility involved in the task is fairly limited. Thus, our second key finding is: *Naively scaling dataset size and diversity does*

---

[4]To the degree of our ability to measure it. Moreover, we make no biological plausibility claims in this work. We are simply motivated by the broad generalization capabilities of a biological visual cortex.

*not improve performance uniformly across benchmarks*. We note that this broad evaluation refutes a naive extrapolation of the positive scaling trends observed in prior work on robot manipulation [8].

Our findings reveal a challenge and opportunity for the community – the search for a PVR that is universally dominant (or "foundational") for EAI calls for innovations in architecture, learning paradigm, data engineering, and more. As a first step towards this open problem, we study *adapting* **VC-1** with either task-specific training losses or datasets (via MAE [18]) to specialize **VC-1** for each domain. We find that adapting **VC-1** results in it becoming competitive with or outperforming the *best prior results on all of the benchmarks* in CORTEXBENCH. We highlight that this comparison is particularly unforgiving, since best prior results are highly domain-specific and are not constrained to share any aspect of their design. To our knowledge, **VC-1** (adapted) is the first PVR that is competitive with (or outperforms) state-of-art results on such a diverse set of EAI tasks (Figure 1).

Finally, we conduct proof-of-concept hardware experiments using **VC-1** in a few-shot imitation learning setting with two platforms: a TriFinger robot and a Franka Emika Panda arm. In this real-world setting, we find that **VC-1** and **VC-1** (adapted) substantially outperform pre-existing PVRs like MVP [8]. We will release code for CORTEXBENCH to enable the EAI, robotics, and CV communities to benchmark their own models, and share our pre-trained models (including **VC-1**) that we believe can serve as a starting point for all visuomotor tasks of interest today.

## 2 Related Work

**Pre-trained visual representations (PVRs).** The last few years have seen increasing interest in the self-supervised learning (SSL) of visual representations [18–22]. These algorithms use contrastive [21, 22], distillation-based [19, 20], or reconstructive [18, 23] objectives for training. Recently, a flurry of works have proposed using the vision transformers (ViTs) [24] with masked image modeling [18, 25, 26], which among other benefits reduces the computation time required for pre-training. In this work, we use one such pre-training algorithm (MAE [18]) to explore scaling and adapting pre-trained visual representations.

**PVRs for embodied AI.** Inspired by the advancements in self-supervised learning, recent work has incorporated visual representation learning into the training pipelines for EAI agents [3–9]. Specifically, [6] evaluate several PVRs trained with supervised or self-supervised learning on a range of EAI tasks, demonstrating promising results under a few-shot imitation learning evaluation protocol. [7–9] introduce new methods for pre-training visual representations using egocentric video data, targeting robot manipulation tasks. Similarly, [3–5] use pre-trained visual representations to improve performance on multiple visual navigation tasks. Closely related, [8] demonstrate that MAE pre-training on internet-scale video and image data can produce effective visual representations for robot manipulation tasks. In contrast, our work studies a larger range of embodied AI tasks (collected in CORTEXBENCH) to understand how PVRs can provide a general-purpose foundation for embodied agents and explores in-domain model adaptation for various tasks.

**Language guided foundation models in EAI.** There has also been some recent works in the area of language-guided representation learning for control. [27] trains a ViT with a masked encoding objective on pairs of image frames and text. [28] focuses on self-supervised representation learning for goal-conditioned value functions using language-aligned videos. Additionally, [29, 30] employ open-vocabulary detectors and vision-language models to detect objects in tabletop views. These detections, along with the image and vision-language model instructions, are then used to train a policy. In [31], a multimodal transformer is pretrained on web-scale image-and-text data, and then used with a transformer-based policy for table-top manipulation tasks.

**Scaling model and dataset size.** Several works show that scaling model and dataset size improves performance on vision tasks like image classification [32–34]. In EAI, Radosavovic et al. [8] find that scaling model and data sizes consistently improves downstream policy performance for robot manipulation tasks. Our work is the first to study this question of scaling on a broad range of EAI tasks and refutes a naive extrapolation of the positive scaling trends observed in [8].

**Adapting PVRs.** When and how to adapt PVRs for downstream applications remains an open research question [35–39]. In the context of EAI, [6] and [40] show that naively fine-tuning PVRs with behavior cloning can reduce performance in simulation, and [8] observe minimal gains in real-world manipulation tasks. In large-scale RL settings, [4, 5] show that end-to-end finetuning considerably improves performance for indoor visual navigation. By comparison, [41] find simple

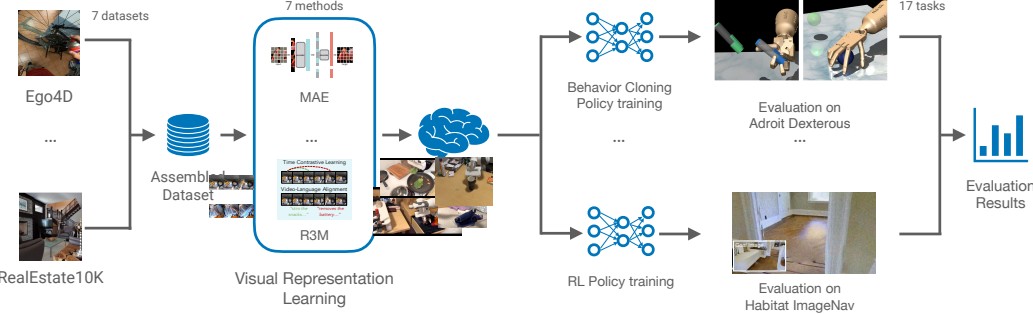

Figure 2: CORTEXBENCH: We systematically evaluate pre-trained visual representations by varying datasets and representation learning algorithms, coupled with reinforcement or imitation learning on diverse EAI tasks.

$k$-nearest-neighbor adaptation works well for real-world visual imitation tasks. Our work neither aims nor expects to be the final word on this fertile topic.

# 3    Benchmarking Progress Towards an Artificial Visual Cortex

We curate CORTEXBENCH (as shown in Figure 1) to evaluate the ability of pre-trained visual representations (PVRs) to support a wide variety of EAI applications. Specifically, we chose 17 diverse tasks drawn from 7 existing EAI benchmarks that have been deemed important by the EAI community. For each task, we delineate a downstream policy learning paradigm (e.g., few-shot imitation learning) and evaluation protocol that follows community standards in each domain (Appendix A.2). By fixing the tasks and downstream learning methods (Figure 2), we are able to focus evaluations on the contribution of PVRs, which in turn allows measuring progress towards the development of an artificial visual cortex for embodied intelligence. We use CORTEXBENCH to conduct the largest and most comprehensive empirical study to-date.

We recommend two evaluation metrics: **Mean Success** and **Mean Rank**. **Mean Success**: the average success rate across all benchmarks. **Mean Rank**: for each benchmark, we rank PVRs based on their success rate; then we average these rankings across all benchmarks.

## 3.1    Embodied AI Tasks in CORTEXBENCH

CORTEXBENCH includes the tasks listed in Table 1, illustrated in Figure 1, and described here:

**Adroit (AD)** [12] is a suite of dexterous manipulation tasks in which an agent must control a 28-DoF anthropomorphic hand to perform a variety of tasks. We study the two hardest tasks from Adroit: `Relocate` and `Reorient-Pen`. In these tasks, an agent must manipulate an object into a goal position and orientation, where the goal must be inferred from the scene.

Table 1: CORTEXBENCH includes tasks from 7 diverse benchmarks with different combinations of observations, actions, goals, and standard policy learning paradigms.

| Benchmark Suite | Observation Space | Action Space | Goal Specification | Policy Learning |
|---|---|---|---|---|
| Adroit (AD) | RGB + proprio. | Continuous | - | IL |
| Metaworld (MW) | RGB + proprio. | Continuous | - | IL |
| DMControl (DMC) | RGB + proprio. | Continuous | - | IL |
| TriFinger (TF) | RGB + proprio. | Continuous | Goal Image/Position | IL |
| ObjectNav (ON) | RGB + proprio. | Discrete | Object Category | IL |
| ImageNav (IN) | RGB | Discrete | Goal Image | RL |
| MobilePick (MP) | RGB + proprio. | Continuous | Goal Position | RL |

**MetaWorld (MW)** [11] is a collection of tasks in which agents command a Sawyer robot arm to manipulate objects in a tabletop environment. We consider five tasks from MetaWorld: `Assembly`, `Bin-Picking`, `Button-Press`, `Drawer-Open`, and `Hammer`, following the evaluations in [7].

**DeepMind Control (DMC)** [10] is a widely studied image-based continuous control benchmark, in which agents perform locomotion and object manipulation tasks. We consider five DMC tasks: `Finger-Spin`, `Reacher-Hard`, `Cheetah-Run`, `Walker-Stand`, and `Walker-Walk`, following [6].

**TriFinger (TF)** is a robot, introduced in [13], composed of a three-finger hand with 3-DoF per finger. We focus on the `Push-Cube` task, which was a part of the Real Robot Challenge 2020 [42]. In this tasks, the agent may use all three fingers to push the cube and move it to a goal location (`Push-Cube`). Additionally, we consider the easier `Reach-Cube` task, which was also studied in [43].

**Habitat** [14] is a simulation platform that includes several visual navigation tasks in which agents explore highly photo-realistic unseen 3D environments. We consider two tasks in Habitat: image-goal navigation (`ImageNav`) [44] and object-goal navigation (`ObjectNav`) [45]. In both, the agent starts at a random location in an unknown 3D environment and must find a goal location – specified with an image from the goal location in `ImageNav` or the name of an object (e.g., 'chair') in `ObjectNav`.

**Habitat 2.0** [15] includes mobile manipulation tasks in which agents control a Fetch robot with a 7-DoF arm, mobile base [46], and suction gripper to rearrange objects in apartment scenes. We consider a challenging version of the `Mobile-Pick` (MP) task from Habitat 2.0, in which an agent must pick up a target object from a cluttered receptacle (e.g., a counter) while starting from a position in which the object is outside of the robot's reach (thus, requiring navigation), using a relaxed dense goal specification (described in Appendix A.3).

**Discussion** CORTEXBENCH encompasses a wide range of tasks, from navigation to manipulation and locomotion. These tasks employ different policy learning methods, including low-level imitation learning (MW, DMC, and TF) and large-scale reinforcement learning (`ImageNav`, `ObjectNav`, and Habitat 2.0). Some tasks require a high-level understanding of semantic scene information (`ImageNav` and `ObjectNav`), while others focus on minor changes in the agent's pose for low-level control (DMC). This diversity in CORTEXBENCH allows us to draw generalized new conclusions about existing and new PVRs.

## 4 Do we already have a visual foundation model for EAI?

First, we evaluate several existing PVRs on CORTEXBENCH to study whether existing open-sourced visual backbones can consistently perform well across all tasks. For all evaluations preceding Section 6, we consider frozen visual representations to disentangle the effect of learned representations from downstream task learning. Specifically, we include the following models:

– CLIP [16] Contrastive image-language pre-training objective; Trained on 400M images-text pairs from the internet (WIT); ViT-B backbone.
– R3M [7] Time-Contrastive video-language alignment pre-training objective; Trained on 5M images from a subset of Ego4D; ResNet-50 backbone.
– MVP [8]. Masked Auto Encoding (MAE) pre-training objective; Trained on 4.5M images from egocentric videos and ImageNet; ViT-B and ViT-L backbones.
– VIP [9]. Goal-conditioned value function pre-training objective; Trained on 5M images from a subset of Ego4D; ResNet-50 backbone.

These models form a representative set for comparisons, spanning different architectures, pre-training objectives and datasets. Additionally, we include randomly initialized ViTs with frozen- and fine-tuned weights to assess the necessity of pre-training and the limitations of pure in-domain learning.

Table 2: Performance of **frozen** pre-trained visual representations (PVRs) on CORTEXBENCH. Best prior results are the best reported in literature prior to this work. Overall, we find that no single PVR consistently performs the best across all benchmarks. However, we find that several of these pre-trained models often outperform a random training from scratch baseline. Best prior results sources (row 1): Adroit and MetaWorld approximated from [7], DMControl from [6], ImageNav from [4], ObjectNav from [47]. Frozen PVR Sources (row 2): Adroit, MetaWorld, and DMControl are the same as SOTA, ImageNav from [4], ObjectNav from [48].

| # | Model | Imitation Learning | | | | | Reinforcement Learning | | Mean | |
| | | Adroit | MetaWorld | DMControl | TriFinger | ObjectNav | ImageNav | Mobile Pick | Rank | Success |
|---|---|---|---|---|---|---|---|---|---|---|
| 1 | Best prior result (any setting) | 75 | 80 | 77 | - | 70.4 | 82.0 | - | | |
| 2 | Best prior result (Frozen PVR) | 75 | 80 | 77 | | 54.4 | 61.8 | - | | |
| 3 | Random (ViT-B) Frozen | $2.0 \pm 2.0$ | $0.5 \pm 0.5$ | $10.1 \pm 0.6$ | $57.8 \pm 0.5$ | $19.2 \pm 0.9$ | $42.1 \pm 0.8$ | $10.8 \pm 1.4$ | 7.2 | 20.4 |
| 4 | Random (ViT-L) Frozen | $2.7 \pm 1.8$ | $0.5 \pm 0.5$ | $9.1 \pm 0.2$ | $57.2 \pm 0.9$ | $19.3 \pm 0.9$ | $45.2 \pm 0.8$ | $20.6 \pm 1.8$ | 6.9 | 22.1 |
| 5 | Random (ViT-B) Fine-tuned | $44.0 \pm 2.0$ | $49.9 \pm 7.3$ | $43.5 \pm 2.4$ | $56.1 \pm 1.3$ | $28.5 \pm 1.0$ | $62.5 \pm 0.7$ | $47.6 \pm 2.2$ | 5.3 | 47.4 |
| 6 | MVP (ViT-B) | $48.0 \pm 3.3$ | $91.2 \pm 2.9$ | $65.9 \pm 2.4$ | $59.7 \pm 0.3$ | $51.2 \pm 1.1$ | $64.7 \pm 0.7$ | $56.0 \pm 2.2$ | 3.1 | 62.4 |
| 7 | MVP (ViT-L) | $53.3 \pm 4.1$ | $87.5 \pm 3.4$ | $69.2 \pm 1.5$ | $74.1 \pm 0.3$ | $55.0 \pm 1.1$ | $68.1 \pm 0.7$ | $65.4 \pm 2.1$ | 2.1 | 67.5 |
| 8 | CLIP (ViT-B) | $47.3 \pm 3.0$ | $75.5 \pm 3.4$ | $55.5 \pm 1.4$ | $62.0 \pm 0.5$ | $56.6 \pm 1.1$ | $52.2 \pm 0.8$ | $49.8 \pm 2.2$ | 3.9 | 57.0 |
| 9 | VIP (RN-50) | $54.0 \pm 4.8$ | $90.1 \pm 2.2$ | $72.5 \pm 2.7$ | $66.7 \pm 0.2$ | $26.4 \pm 1.0$ | $48.8 \pm 0.8$ | $7.2 \pm 1.2$ | 4.0 | 52.3 |
| 10 | R3M (RN-50) | $73.3 \pm 2.0$ | $96.0 \pm 1.1$ | $81.1 \pm 0.7$ | $69.2 \pm 0.8$ | $22.7 \pm 0.9$ | $30.6 \pm 0.7$ | $33.2 \pm 2.1$ | 3.4 | 58.0 |

Table 2 shows the evaluation results aggregated by benchmark; no single model excels in all cases. Among all of the models evaluated, R3M performs the best on Adroit, MetaWorld, and DMControl. While MVP (ViT-L) performs best on TriFinger, ImageNav, and Mobile Pick. CLIP, on the other hand, achieves the best results on ObjectNav. The variance in performance of existing PVRs on CORTEXBENCH is further illustrated in Figure 5 in Appendix A.4 and highlights that we do not yet have a single, strong performing artificial visual cortex for EAI.

## 5 Analyzing the Scaling Hypothesis for EAI

The previous section investigated models pre-trained on datasets of varying size and diversity. Interestingly, while the model pre-trained on the largest dataset (CLIP) performs well on one benchmark (ObjectNav) it does not perform well across all tasks. We now ask: *how much does the relevance and*

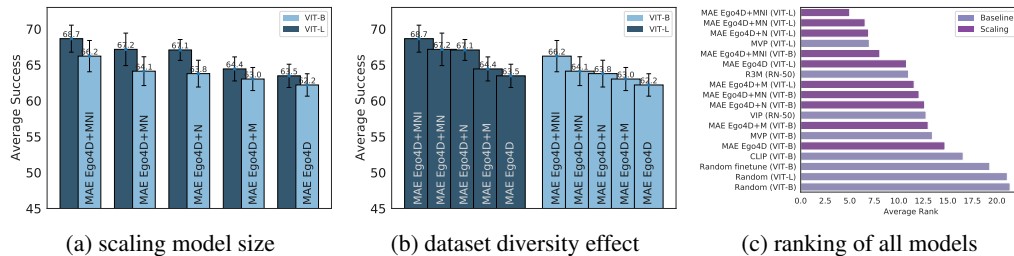

| (a) scaling model size | (b) dataset diversity effect | (c) ranking of all models |

Figure 3: Visualization of scaling hypothesis model performance averaged over CORTEXBENCH. We see modest but positive scaling trends in both (a) scaling model size and (b) dataset diversity. (c) Average ranking of existing PVRs (Table 2) and scaling models (Table 4); **VC-1**: **Ego4D+MNI** (ViT-L) has the highest average rank.

*diversity of the pre-training dataset and the model size matter?* To study this, we fix the pre-training objective (MAE [18]) and vary the composition of the pre-training dataset and the size of the visual backbone (ViT-B with 86M and ViT-L with 307M parameters). We measure the corresponding changes in performance on CORTEXBENCH. MAE is selected for these experiments due to the strong performance on CORTEXBENCH of MVP [8] (Table 2), which uses the MAE pre-training objective.

## 5.1 Constructing a Pre-training Dataset for EAI

To evaluate the impact of dataset size and diversity on CORTEXBENCH, which involve navigation and manipulation tasks, we employ a combination of 7 datasets. One cluster of datasets – Ego4D [49], 100 Days of Hands (100DOH) [50], Something-Something v2 (SS-V2) [51], and Epic Kitchens [52] – contains videos of people manipulating objects and is comparable to the datasets used in MVP [8]. A second cluster consists of egocentric indoor navigation datasets: the Real Estate 10K dataset [53] and the Open-

Table 3: Datasets assembled to study effects of pre-training dataset size, diversity, and relevance – the largest (**Ego4D+MNI**) has 5.6M frames. More details in A.5.

| Name | Frames Used |
| --- | --- |
| **Ego4D** | 2,790,520 |
| **Ego4D+M** (Manip) | 3,538,291 |
| **Ego4D+N** (Nav) | 3,593,049 |
| **Ego4D+MN** (Manip, Nav) | 4,340,820 |
| **Ego4D+MNI** (Manip, Nav, ImageNet) | 5,621,987 |

House24 dataset (described in Appendix A.5.1). Finally, we include ImageNet [54]. We strategically select dataset combinations (shown in Table 3) to answer the following questions:

– What is the impact of scaling dataset size and diversity?
– How do *less-relevant* datasets influence the performance of PVRs on EAI tasks?

**Ego4D** [49] (our base dataset) includes a wide range of egocentric videos consisting of *daily life activities* such as home, leisure, transportation, and workplace activities.

**Ego4D+M** extends **Ego4D** with videos from 100DOH, SS-v2, and Epic Kitchens, resulting in 3.5M frames and making this dataset primarily focused on object manipulation scenarios.[5]

**Ego4D+N** extends **Ego4D** with two indoor navigation datasets: OpenHouse24 and RealEstate10K. This dataset is similar in size to **Ego4D+M** (3.5M frames) but is more diverse because it contains a larger proportion of navigation data than the manipulation-centric datasets **Ego4D** and **Ego4D+M**.

**Ego4D+MN** combines **Ego4D** with the three object manipulation-centric datasets and two indoor navigation dataset, resulting a dataset with 4.3M frames. While larger than **Ego4D+M** and **Ego4D+N**, it does not include any new types of data beyond the manipulation and navigation videos in the previous subsets. Thus, it is no more diverse than **Ego4D+N** (which includes both types of data).

**Ego4D+MNI** includes **Ego4D**, all manipulation and navigation datasets, and ImageNet for a total of 5.6M frames. This dataset is used to study the role of internet images for our benchmark tasks.

## 5.2 Scaling Hypothesis Findings

We now turn to analyzing the effect of increasing model size, dataset size, and dataset diversity. The full set of results is shown in Figure 3 and Table 4. The key takeaways are:

**Model Size.** We find that increasing model size positively impacts performance on CORTEXBENCH. Specifically, in Figure 3a, we find that with all pre-training datasets, switching from ViT-B to ViT-L

---

[5]While **Ego4D** does contain navigation data, it is heavily skewed towards object manipulation.

Table 4: Performance of scaling hypothesis models on CORTEXBENCH. We find that on average the **VC-1 EGO4D+MNI** (VIT-L) model performs best, but is not the best for each benchmark.

| # | Model | Adroit | Meta-World | DMControl | TriFinger | ObjectNav | ImageNav | Mobile Pick | Mean Rank | Mean Success |
|---|-------|--------|-----------|-----------|-----------|-----------|----------|-------------|-----------|--------------|
| 1 | Best prior result (any setting) | 75 | 80 | 77 | - | 70.4 | 82.0 | - | | |
| 2 | Rand (ViT-B) fine-tuned | 44.0 | 49.9 | 34.2 | 55.0 | 28.5 | 65.0 | 47.6 | | |
| 3 | Best result Table 2 (Frozen PVR) | 73.3 | 96.0 | 81.1 | 74.1 | 56.6 | 68.1 | 65.4 | | |
| 4 | Ego4D (VIT-B) | 48.7 ± 1.3 | 86.1 ± 2.1 | 64.1 ± 2.3 | 68.3 ± 1.1 | 46.8 ± 1.1 | 64.0 ± 0.7 | 57.4 ± 2.2 | 8.6 | 62.2 |
| 5 | Ego4D (VIT-L) | 50.0 ± 1.2 | 92.9 ± 2.4 | 60.8 ± 3.3 | 69.7 ± 0.5 | 47.6 ± 1.1 | 55.8 ± 0.8 | 67.6 ± 2.1 | 5.9 | 63.5 |
| 6 | Ego4D+N (VIT-B) | 50.0 ± 2.4 | 86.4 ± 2.9 | 59.5 ± 2.4 | 67.8 ± 1.3 | 54.7 ± 1.1 | 68.7 ± 0.7 | 59.4 ± 2.2 | 7.2 | 63.8 |
| 7 | Ego4D+N (VIT-L) | 54.0 ± 1.2 | 89.1 ± 2.9 | 66.4 ± 1.7 | 66.9 ± 0.4 | 57.4 ± 1.1 | 70.5 ± 0.7 | 65.2 ± 2.1 | 3.5 | 67.1 |
| 8 | Ego4D+M (VIT-B) | 51.3 ± 2.4 | 83.5 ± 2.6 | 64.3 ± 1.8 | 69.1 ± 0.4 | 47.3 ± 1.1 | 65.8 ± 0.7 | 59.8 ± 2.2 | 7.0 | 63.0 |
| 9 | Ego4D+M (VIT-L) | 52.0 ± 1.3 | 88.3 ± 3.2 | 64.7 ± 2.4 | 64.7 ± 0.9 | 47.3 ± 1.1 | 65.5 ± 0.7 | 68.6 ± 2.1 | 6.0 | 64.4 |
| 10 | Ego4D+MN (VIT-B) | 48.7 ± 2.4 | 85.3 ± 5.2 | 64.2 ± 1.9 | 70.3 ± 0.5 | 52.8 ± 1.1 | 68.9 ± 0.7 | 58.6 ± 2.2 | 6.9 | 64.1 |
| 11 | Ego4D+MN (VIT-L) | 52.7 ± 4.2 | 86.7 ± 3.9 | 69.7 ± 3.3 | 72.4 ± 0.5 | 58.4 ± 1.1 | 69.1 ± 0.7 | 61.2 ± 2.2 | 3.1 | 67.2 |
| 12 | Ego4D+MNI (VIT-B) | 54.0 ± 4.0 | 89.6 ± 3.9 | 63.8 ± 2.7 | 72.2 ± 0.6 | 55.4 ± 1.1 | 67.9 ± 0.7 | 60.6 ± 2.2 | 4.4 | 66.2 |
| 11 | **VC-1**: Ego4D + MNI (VIT-L) | 59.3 ± 5.2 | 88.8 ± 2.2 | 66.9 ± 1.4 | 71.7 ± 0.4 | 60.3 ± 1.1 | 70.3 ± 0.7 | 63.2 ± 2.2 | 2.4 | 68.7 |

improves average performance on CORTEXBENCH. However, in Table 4, we find exceptions where this general trend does not hold. For instance, when pre-trained on **Ego4D+MNI**, the ViT-B model outperforms the ViT-L model on MetaWorld and TriFinger.

**Dataset Size and Diversity.** Figure 3b shows that, in general, increasing dataset size and diversity leads to improved performance. Models are are ordered from right to left by increasing size and the diversity of their pre-training dataset, and we mostly see improvements for both ViT-B and ViT-L. For instance, **Ego4D+M** slightly improves upon **Ego4D** by 0.6 and 0.9 points (62.2 → 62.8 and 63.5 → 64.4) in the case of ViT-B and ViT-L, respectively. The gains with **Ego4D+N** are larger and it outperforms **Ego4D** by 1.6 points using ViT-B (62.2 → 63.8) and by 3.6 points for ViT-L (63.5 → 67.1). It is interesting to note that **Ego4D+N** has a larger improvement over the base **Ego4D** dataset than **Ego4D+M**, even though **Ego4D+N** and **Ego4D+M** dataset are similar in size. In these results, we find that increasing diversity by adding indoor navigation data improves performance more than adding additional manipulation data to **Ego4D**.

Additionally, we find that pre-training on **Ego4D+MN** is roughly on par with pre-training on **Ego4D+N**. We see a 0.3 and 0.1 point difference (63.8 → 64.1 and 67.1 → 67.2) for ViT-B and ViT-L, respectively, even though **Ego4D+MN** has about 800K more training frames. Together with the results from above this demonstrates that increasing data diversity seems to matter more than simply increasing dataset size.

Next, we find that adding ImageNet positively impacts average performance on CORTEXBENCH. For example, models pre-trained on **Ego4D+MNI** outperform those pre-trained on **Ego4D+MN** by 1.9 points (64.1 → 66.2) for ViT-B and 1.5 points (67.2 → 68.7) for ViT-L. Interestingly, these results demonstrate that including static internet images can significantly boost performance on EAI tasks. This finding further highlights the importance of seeking data diversity to build better representations.

Finally, our largest model (ViT-L) pre-trained on all datasets (**Ego4D+MNI**), achieves the best rank when averaged across all benchmark tasks (Table 4 row 11), with a mean rank of 2.4. We call this model **VC-1**, and will open-source it. **VC-1** is superior to the second-best model (**Ego4D+MN** ViT-L, Table 4 row 9), which has an average rank of 3.1.

However, upon further dis-aggregation, we observe we find that while **VC-1** performs best on average, it is not the best for each benchmark. For example, the best model for Mobile Pick, a mobile manipulation task, is a ViT-L trained on **Ego4D+M** and the best model for ImageNav, an indoor navigation task, is the ViT-L trained on **Ego4D+N**. These findings suggest that task-specific pre-training datasets could enhance the performance of models on individual tasks. However, it is important to note that this approach would lead to multiple pre-trained models, each tailored to a specific task, and not a unified visual foundation model.

### 5.3 How does VC-1 compare to existing PVRs?

We now compare **VC-1** to PVRs from Section 4. On average, **VC-1** has the best rank across all benchmarks (Figure 3c). In terms of mean success, **VC-1** (Table 4 row 11) outperforms MVP (ViT-L) by +1.2 points (67.5 → 68.7), R3M by +10.7 (58.0 → 68.7), CLIP by +11.7 (57.0 → 68.7), and end-to-end fine-tuning from scratch +19.6 (49.1 → 68.7).

Impressively, **VC-1** outperforms CLIP *on every benchmark* (Figure 4), despite training on a 70X smaller dataset, emphasizing the importance of egocentric interaction datasets. **VC-1** also outperforms fine-tuning from scratch on every benchmark, indicating that PVRs trained with out-of-domain data can outperform in-domain, end-to-end learning.

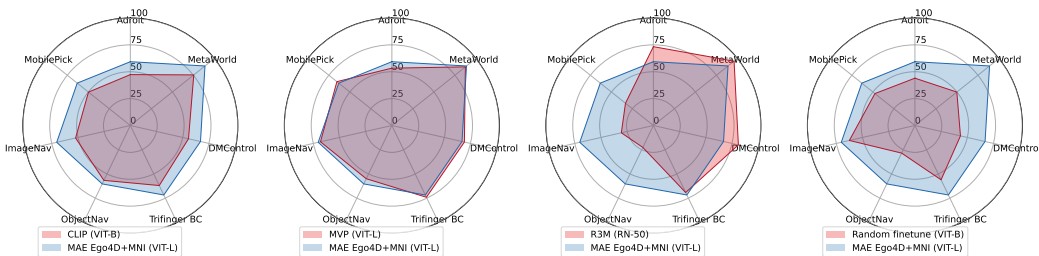

Figure 4: Comparison of **VC-1** with existing PVRs. **VC-1** matches or exceeds existing PVRs on all benchmarks except R3M on AD, MW, and DMC, indicating an opportunity for model adaptation.

The MVP model is the most similar in terms of results, architecture, and pre-training objective to **VC-1**, with the main difference being the addition of a *convolutional stem* in MVP. **VC-1** outperforms MVP VIT-L by 1.3 points on mean success and performs better on four out of seven benchmarks (Figure 4), likely due to the use of a more diverse dataset.

When compared to R3M, **VC-1** demonstrates superior performance both on average and in 4 out of 7 benchmarks (Figure 4). However, R3M outperforms **VC-1** on Adroit, MetaWorld and DMControl benchmarks. The cause of this gap is unclear – it could be due to differences in pre-training objectives, datasets, or the backbone architecture. That said, the fact that R3M, which uses a ResNet-based architecture, performs better on some tasks than **VC-1** which employs a Transformer-based backbone, suggests the potential value of exploring new architectures that integrate the strengths of both approaches - inductive biases and scalability. Taken together, these observations highlight the need for more robust and standardized evaluations on benchmarks like CORTEXBENCH.

Overall, **VC-1** is effective across a broad set of tasks and thus a reasonable starting point for novel EAI problems. However, it is not always the best model for a specific task. This leads us to theorize that there is a domain gap that might be bridged with dataset engineering or adaptation of the PVR.

## 6 Adapting VC-1

In prior sections, we focused on evaluating **VC-1** as a **frozen** PVR. We now study if *adapting* **VC-1** can improve results in downstream tasks. We use a broad definition of adaptation [55], which, in the context of large pre-trained foundation models, can take several forms from simple prompting [56], to selectively updating some or all weights of the backbone [5, 35, 57].

In the context of PVRs for EAI, adaptation can serve at least two purposes. The first is *task-specialization* in the feature extraction stage. Since **VC-1** was trained with MAE [18], it captures features that are generally useful for reconstructing images (see **VC-1** attention maps in Appendix A.10). Adaptation can specialize the visual backbone to extract features required for performing specific EAI tasks. Secondly, adaptation can also help *mitigate domain-gap* that might exist between pre-training and evaluation settings. In general, domain-gap can arise for several reasons such as poor coverage in pre-training datasets or deployment in novel conditions (e.g., on robots) not seen in the pre-training data (e.g., in human-centric video datasets). Domain gap is naturally instantiated in our setup, since **VC-1** was pre-trained on real-world, human video data while our downstream evaluation in CORTEXBENCH uses simulated EAI domains with different visual characteristics.

In this work, we explore two methods for adaptation: end-to-end (E2E) fine-tuning and MAE adaptation. While we do not explore prompting-based adaptation, because our visual encoders were not originally designed to utilize prompts, it may be an interesting direction for future work.

**End-to-end (E2E) fine-tuning** with a task-specific loss function can in-principle capture both of the aforementioned benefits of adaptation, and is widely used in computer vision literature [18, 20, 58, 59]. To study E2E fine-tuning of **VC-1**, we use the same policy learning methods described in Appendix A.2, except we allow updates to the **VC-1** weights. In the CORTEXBENCH results in Table 5, we find an interesting mixed result. In domains that involve large-scale IL or RL (ObjectNav, ImageNav, and Mobile Pick), the strategy proposed in [5] of adapting **VC-1** with E2E fine-tuning significantly improves performance over using a frozen **VC-1** backbone. Specifically, we see an improvement in ObjectNav success rate (SR) of +7.4 (60.3 → 67.7), ImageNav SR of +11.3 (70.3 → 81.6), and Mobile Pick SR of +10.8 (63.2 → 74.0). These results suggest that E2E

Table 5: Adapting **VC-1** with end-to-end fine-tuning or self-supervised learning (MAE) on task-specific data leads to substantial gains. Best prior results are across any setting (frozen/finetuned). Best result (our exp.) are from Section 5. ** indicates that, on hardware, we only evaluated MVP (ViT-L), the closest competitor to **VC-1**. The TF Push-Cube results are averaged across 12 trials with randomized start/goal configurations. The Franka results are averaged across 4 tasks and 10 trials per task (details in Appendix A.12).

| | | CortexBench | | | | | | | Hardware | |
|---|---|---|---|---|---|---|---|---|---|---|
| # | Method | Adroit | MetaWorld | DMControl | TriFinger | ObjectNav | ImageNav | Mobile Pick | TF Push-Cube | Franka |
| 1 | Best prior result | 75 | 80 | 77 | - | 70.4 | 82.0 | - | - | - |
| 2 | Best result (our exp.) | 73.3 ± 2.0 | 96.0 ± 1.1 | 81.1 ± 0.7 | 74.1 ± 0.3 | 60.3 ± 1.1 | 70.5 ± 0.7 | 68.6 ± 2.1 | 31.9 ± 4.3** | 55.0 ± 10.4 ** |
| 3 | In-domain MAE (baseline) | 47.3 | 83.4 | 77.6 | 80.4 ± 0.32 | 39.9 ± 1.09 | 47.6 ± 0.77 | 51.6 ± 2.23 | 22.9 ± 5.4 | 35.0 ± 13.9 |
| 4 | **VC-1** | 59.3 ± 5.2 | 88.8 ± 2.2 | 66.9 ± 1.4 | 71.7 ± 0.4 | 60.3 ± 1.1 | 70.3 ± 0.7 | 63.2 ± 2.2 | 45.8 ± 6.5 | 70.0 ± 10.9 |
| 5 | **VC-1** E2E fine-tuning | 15.9 ± 9.8 | 22.7 ± 9.7 | 6.7 ± 3.8 | 70.9 ± 0.5 | 67.7 ± 1.05 | 81.6 ± 0.6 | 74.0 ± 1.96 | 52.7 ± 9.2 | 67.5 ± 13.4 |
| 6 | **VC-1** MAE adaptation | 72.0 ± 3.3 | 96.0 ± 1.8 | 80.9 ± 1.8 | 80.6 ± 0.25 | 57.4 ± 1.11 | 67.0 ± 0.73 | 62.4 ± 2.16 | 43.5 ± 6.7 | 85.0 ± 9.1 |

fine-tuning of **VC-1** can achieve the benefits of both task-specialization and domain adaptation. A comparison of **VC-1** attention maps before and after adaptation is provided in Appendix A.10.

However, in few-shot IL domains (Adroit, MetaWorld, DMC, and TriFinger), E2E fine-tuning does not result in improvements. In fact, in these domains, it leads to a drop in performance, a finding consistent with prior work [6, 40]. We hypothesize that the poor performance of E2E fine-tuning in few-shot IL domains is caused by overfitting, due to fine-tuning a large model with 307M parameters on a small dataset ($\leq 50K$ frames).

**MAE adaptation to mitigate domain-gap.** As an alternative to E2E fine-tuning, we explore adapting **VC-1** with self-supervised learning (SSL). Specifically, in MAE adaptation we continue training the backbone network with the MAE [18] pre-training objective on task-specific data. Then, we freeze these adapted representations and use them to learn task-specific policies. We note that in MAE adaptation, the backbone is adapted using the same data that is used for training the policy (e.g., frames from expert demonstrations), and no additional in-domain datasets are used. While this adaptation strategy cannot address task-specialization, it may serve to mitigate domain gap.

For MAE adaptation, we initialize with **VC-1** weights, and then train with MAE for 100 epochs. In domains where expert demonstrations are available (i.e., Adroit, MetaWorld, DMControl, TriFinger, and ObjectNav), we use the RGB frames from these demonstrations for adaptation. In the remaining two benchmarks (ImageNav and Mobile Pick) we sample frames from training environments to create adaptation datasets. Finally, to isolate the importance of initializing with **VC-1** weights, we train in-domain MAE baselines by starting from a random initialization and then following the same approach used for MAE adaptation.

In the CORTEXBENCH results in Table 5, we observe MAE adaptation substantially improves performance in few-shot learning domains. Specifically, on Adroit performance improves by +12.7 (59.3 → 72.0), MetaWorld by +7.2 (88.8 → 96.0), DMC by +14.0 (66.9 → 80.9), TriFinger by +8.9 (71.7 → 80.6). Interestingly, in DMC and TriFinger, the in-domain MAE baseline (row 3) performs surprisingly well, highlighting the importance of in-domain data for representation learning. Finally, in large-scale IL or RL domains (ObjectNav, ImageNav, and Mobile Pick), we find MAE adaptation results in small reductions in performance from **VC-1** (row 4 vs. 6). In these domains, where substantial amounts of data is available for task-specific training (large-scale IL or RL), we find that E2E fine-tuning is the superior approach for adaptation. In aggregate, these results suggests that MAE adaptation can be explored as a powerful alternative in few-shot domains or where E2E fine-tuning fails.

Overall, we find that *adapting* the **VC-1** model leads to performance improvement in all the benchmark domains. Furthermore, on MetaWorld, DMControl, and TriFinger, **VC-1** with MAE adaptation (row 6) is comparable with the best known results (SoTA) and the best results from previous sections (rows 1 and 2). Similarly, on ImageNav and Mobile Pick, **VC-1** with E2E fine-tuning (row 5) matches or exceeds the best results. Together, these results demonstrate that **adaptation** of PVRs can be a powerful paradigm for EAI, especially when compared to training representations from scratch.

## 7 Proof-of-Concept Hardware Experiments

In addition to simulation experiments with CORTEXBENCH, we also explore proof-of-concept hardware experiments utilizing **VC-1** as a backbone PVR for training policies with IL. Our hardware evaluation spans two platforms: TriFinger (1 task) and a Franka-Emika Panda arm (4 tasks). Setup details are provided in Appendix A.11 and A.12. We follow a similar experiment protocol to the simulated counterparts by studying few-shot imitation learning, where the demonstrations are collected directly in the real-world via tele-operation or hand-designed controllers.

We study the cases when using **VC-1** as a frozen PVR, **VC-1** with MAE adaptation and **VC-1** with E2E adaptation (as specified in Section 6), and the MVP model as a baseline (best PVR from Section 4). The results are summarized in the hardware section of Table 5. Overall, we observe similar trends to our findings in Section 6. We observe that in frozen mode (row 4), **VC-1** substantially outperforms both MVP (row 2) and in-domain MAE (row 3) in both robot setups. We also find that adaptation via E2E fine-tuning improves real-world TriFinger performance and that MAE adaptation leads to large improvements on Franka tasks. We note that while MAE adaptation does not help TriFinger performance, this is likely due to the mere 300 real robot images available for adaptation. Together, these results suggest that learning task-specific features (via fine-tuning) was more important than closing any domain gap (via MAE adaptation) for TriFinger. Overall, these results demonstrate that **VC-1** can effectively function as a PVR for multiple hardware platforms, and can outperform prior PVRs that have shown success on hardware, such as MVP. Furthermore, it reinforces finding from Section 6 that adapting **VC-1** with SSL objectives (MAE) can improve the performance.

## 8 Discussion

This work introduced CORTEXBENCH, which comprises 17 different EAI task spanning locomotion, indoor navigation, and dexterous and mobile manipulation; and conducted the most comprehensive study to-date of PVRs (or visual foundation models) for EAI. We find that (1) despite significant progress in a number of narrow domains, we do not yet have a universal visual backbone for all EAI tasks of interest, (2) naively scaling model size and pre-training data diversity does not improve performance universally across all tasks, but does so on average, (3) adapting our largest pre-trained model (**VC-1**) results in performance that is *competitive with or outperforms the best known results on all benchmarks* in CORTEXBENCH, and (4) **VC-1** and adaptation show proof-of-concept generalization to hardware. Our study is an attempt to unify various EAI areas using the perception module as a cornerstone. The study and development of visual representations for sensorimotor control today appears splintered across different sub-communities studying egocentric computer vision, locomotion, navigation, dexterous and mobile manipulation. However, we contend that this cannot be the final solution. Biological organisms have one visual cortex, not one per 'task'. Analogously, it must be possible for an embodied AI agent to have one universal artificial visual cortex supporting a diverse range of sensorimotor skills, environments, and embodiments. We speculate that learning visual representation using temporal signals, 3D spatial priors, or objectness will help make progress towards this goal. Our final contention is that in order for the research community to develop such a model, we need to create benchmarks that test broad generalization capabilities; we hope CORTEXBENCH will help the community make progress towards that.

## Acknowledgements

The Georgia Tech effort was supported in part by ONR YIP and ARO PECASE. The views and conclusions contained herein are those of the authors and should not be interpreted as necessarily representing the official policies or endorsements, either expressed or implied, of the U.S. Government, or any sponsor.

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

# A  Appendix

## A.1  Limitations

This study presents a thorough examination of visual foundation models but has several limitations. Firstly, in proposing a benchmark, we sought to find a balance between task diversity and the computational resources required for evaluation. However, new and challenging benchmarks in embodied AI, such as those presented in [60], continue to emerge and may merit inclusion in future studies to track progress in this field. Additionally, while we have focused on masked auto-encoders (MAE) as the pre-training objective and ViTs as the architecture in our study, there may be other SSL algorithms that exhibit different scaling behaviors or superior performance on the proposed tasks in our benchmark. Lastly, the adaptation procedures studied in this work necessitate separate training on in-domain datasets, as well as careful tuning of hyperparameters such as the number of training epochs and sampling ratio of the dataset. This results in a significant effort to produce a separate adapted PVR model for each benchmark evaluated on our benchmark, and the overall effort increases proportionately with the number of benchmarks included in the study.

In conclusion, it is important to note that although we utilize real-world images and videos for pre-training our visual representation models (PVRs), the evaluation benchmarks used in this study serve as proxies for actual robotic tasks, and thus, the performance of the PVR models on real robots may differ from the rankings established in this study. Further research is necessary to fully evaluate the effectiveness of these models in real-world scenarios.

## A.2  Overview of Downstream Policy Learning in CORTEXBENCH

Given a frozen PVR, an agent needs to learn a policy for each task. The EAI community has developed a range of policy learning algorithms from few-shot imitation learning (IL) to large-scale reinforcement learning (RL). For each task in CORTEXBENCH, we conform to the community standard for achieving state-of-art performance in that domain.

**"MuJoCo Tasks"** On the tasks from the Adroit, MetaWorld, and DMC suites we train policies using behavior cloning on a small number of expert demonstrations (100 for Adroit and DMC and 25 for MetaWorld), which follows [6, 7]. Specifically, we train policies for 100 epochs and report the average rollout performance on the test set for the best intermediate policy during training. For all tasks, we use frame-stacking and a 3-layer MLP policy network. When using vision transformers (ViT) based PVRs, we use the `[CLS]` token as input to the policy, and with ResNets we use features from the final convolutional layer after global average pooling. These design choices follow prior work such as [7, 8].

**"TriFinger Tasks"** For TriFinger, we train policies using behavior cloning on 100 demonstrations per task. Specifically, we train a policy network composed of a 3-layer MLP for 100 epochs for `Reach-Cube` and 1,000 epochs for `Push-Cube`. We report the average score for the best checkpoint over the course of training. The input to the policy is the `[CLS]` token for ViT-based PVRs and average pooled features from the last convolutional layer for ResNet-based models.

**"Habitat Tasks"** We train `ObjectNav` policies with behavior cloning on 77k human demonstrations [61] collected by [62], totaling 360M environment steps. For `ImageNav` and `Mobile-Pick`, we use RL for 500M environment steps with DD-PPO [63] and VER [36]. We use patch representations for ViT-based PVRs and grid-features from last convolutional layer for ResNet models, passed through a compression layer [14] for a lower dimensional representation for use by the policy layers, which is a 2-layer LSTM for navigation and a 2-layer GRU for manipulation.

## A.3  Additional Details of Tasks and Downstream Learning in CORTEXBENCH

In this section, we discuss further details for a subset of downstream tasks in CORTEXBENCH.

**ImageNav.** Our study conducts `ImageNav` experiments using the standard dataset presented in [64]. This benchmark utilizes the Habitat simulator [15, 65] and is situated within the Gibson [66] environments, which comprise 72 training scenes and 14 validation scenes. The validation set includes 300 episodes for each scene, for a total of 4,200 episodes. In this benchmark, agents are modeled as cylinders with a height of 1.5m, radius of 0.1m, and sensors located 1.25m above the center of the

base. The RGB camera has a resolution of 128×128 and a 90° field-of-view. Agent is able to take up to 1000 steps within the environment and are deemed successful if they reach a location within 1m of the goal position and call STOPACTION.

To train the agents within the Gibson environments, we utilize 500M timesteps (25k updates) with 320 environments running in parallel. Each environment collects up to 64 frames of experience, which is followed by 2 PPO epochs utilizing 2 mini-batches. Unless otherwise specified, we use a learning rate of $2.5 \times 10^{-4}$ for training the agents and update the parameters using the AdamW optimizer with a weight decay of $10^{-6}$. We train agents with the reward functions presented in [67] utilizing the following settings: success weighting $c_s = 5.0$, angle success weighting $c_a = 5.0$, goal radius $r_g = 1.0$, angle threshold $\theta_g = 25°$, and slack penalty $\gamma = 0.01$. We evaluate performance every 25M steps of training and report metrics based on the highest success rate (SR) achieved on the validation set.

**ObjectNav.** We present an evaluation of object navigation (`ObjectNav`) using the HM3D-SEM dataset [61]. The dataset is comprised of 80 training, 20 validation, and 20 testing scenes and utilizes the Habitat simulator [15, 65] and HM3D [68] environments. Our results are reported on the v0.1 HM3D-SEM VAL split, which was used in the 2022 Habitat Challenge [69] `ObjectNav` benchmark. The agent in this evaluation is modeled after the LocoBot [70] with a height of 0.88m, radius of 0.18m, and sensors placed at the top of the agent's head. The RGB camera has a 640×480 resolution and a 79° horizontal field of view. The task for the agent is to locate objects from one of 6 categories: *'chair'*, *'bed'*, *'plant'*, *'toilet'*, *'tv/monitor'*, and *'sofa'* within 500 steps. Successful episodes are determined by the agent stopping within 0.1m of a viewpoint that is (a) within 1m of any instance of the target object and (b) from which the object is visible, as outlined in the evaluation protocol of [45].

We utilize a dataset of human demonstrations for training our imitation learning agent in the task of `ObjectNav`. The dataset was collected using Habitat-Web [61, 71] and Amazon Mechanical Turk, and consists of $77k$ demonstrations for 80 scenes from the HM3D-SEM dataset [69]. Each scene contains approximately 158 episodes, each with a unique goal object category and a randomly set start location, resulting in approximately 950 demonstrations per scene. The dataset includes a total of $\sim$12.1 million steps of experience, with an average of $\sim$159 steps per episode. By leveraging this human demonstration data, our imitation learning agent is able to learn a more effective policy for navigating to objects in complex environments.

We trained object navigation (`ObjectNav`) agent in the HM3D environment for an approximate total of 400 million steps, utilizing 25,000 updates and 512 parallel environments. Similar to our previous image-based navigation (`ImageNav`) experiments, we employed a weight decay of $10^{-6}$ and utilized different learning rates for the visual encoder and other elements of the model. Specifically, we used a learning rate of $10^{-4}$ for the visual encoder and $10^{-3}$ for all other elements, with the AdamW optimizer. To ensure the quality of our trained models, we evaluated checkpoints after every $10M$ steps and only reported metrics for the checkpoints with the highest validation success rate.

**Mobile Pick.** We investigate the Habitat 2.0 Rearrangement task proposed by [15]. This task involves a mobile manipulation scenario in which a Fetch robot navigates an ReplicaCAD apartment to pick up a target object from a cluttered receptacle using a mobile base [46]. The robot starts from a non-trivial position and must utilize a variety of sensors, including an egocentric RGB camera, proprioceptive joint sensing, and an object grasping indicator. The action space for the robot includes continuous control over the robot's 7-DOF arm, base movement, and suction gripper. We relax the dense goal specification, where the relative position between the end-effector and the target object must be updated at each step, to a sparse goal specification, where this information is only provided at the start of the episode. This relaxation places greater emphasis on visual input and makes the task significantly more challenging.

**TriFinger Tasks.** The TriFinger tasks are implemented in Pybullet. For `Reach-Cube`, the state for the BC policy is $[x_t^{ft}, z_t]$, where $x_t^{ft}$ is the current fingertip position and $z_t$ is the latent visual state vector, obtained by passing the current image observation through the PVR. The success metric captures how close the fingertip is to the optimal distance from the center of the cube, accounting for the half=width of the cube. For `Push-Cube`, the state for the BC policy is $[x_t^{ft}, z_t, \Delta x_g^c]$, where $\Delta x_g^c$ is the goal position for the cube, specified as a displacement from its initial position. Here the success

is the distance of the center of the cube to the target goal position. We train a policy network with hidden layers of size 2000 and learning rate $10^{-4}$ for up to 100 epochs for the reach task and 1000 epochs for the Push-Cube task.

## A.4 Additional Analysis of Existing Pre-Trained Visual Representations (PVRs)

The rank distribution for existing PVRs (shown in Figure 5) demonstrates that there is high variability in the performance of the models (from prior work) across the benchmarks in CORTEXBENCH.

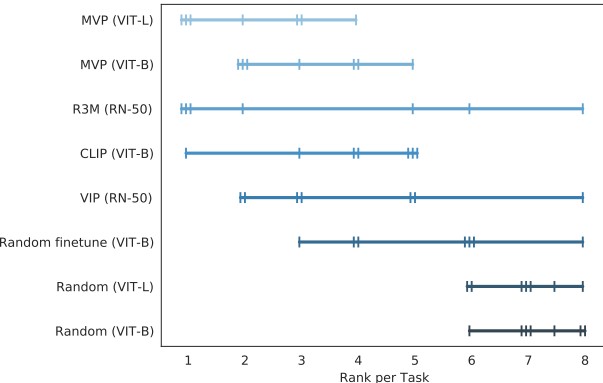

Figure 5: Rank distribution of existing pre-trained visual representations. For every model, we compute the ranks it achieved on each of the 7 benchmarks. We visualize them as vertical lines, where each rank number $x$ receives a tick if that model achieved such rank $x$. For instance, MVP (ViT-L) achieves ranks 1,1,1,2,3,3,4 across the 7 benchmarks. Significant variability exists in the performance of PVRs across benchmarks.

## A.5 Scaling Hypothesis Datasets

The datasets composed for scaling hypothesis experiments are detailed in Table 6.

### A.5.1 OpenHouse24 Dataset

The OpenHouse24 (OH24) dataset is a collection of video walk-throughs of furnished residential real estate properties. Over 1600 homes are represented in the dataset, totaling 139 hours of video footage. Each home is traversed in a continuous shot with a stable HD RGB camera by an operator that efficiently visits each room. The dataset represents a diverse set of properties, including (but not limited to) small and large suburban homes, high-rise apartments, ranch homes, and condos. The ensuing walk-throughs range from under a minute to 14 minutes in length, with the average taking 5 minutes and 12 seconds. The dataset will be open-sourced by a separate research project.

## A.6 Scaling Hypothesis Pretraining Details

To train the MAE models, we use the official codebase released by the authors on GitHub [18] and use the default hyperparameters provided by the repo to train the ViT-B and ViT-L models. We found the default values worked well on the CORTEXBENCH. However, we do vary the number of epochs we use to train the different models in Section 5 given the different dataset sizes. We choose the number of epochs per run such that the number of model updates remain constant across all runs and match the number of model updates taken by MAE on the ImageNet dataset. We provide details about the dataset sizes and the epochs calculated for the different runs in Table 7.

## A.7 Additional Analysis of Scaling Hypothesis Results

Figure 6 (left) provides additional evidence of the significance of pre-training dataset diversity. We observe that while the ViT-L models trained in **Ego4D+M** and **Ego4D+N** datasets, achieve the best result in one of the benchmarks, they perform the worst and second-worst in other benchmarks. However, by adding diversity, the **Ego4D+MN** and **Ego4D+MNI** models have decreased variance

Table 6: Datasets used for our scaling hypothesis experiments, containing up to 5.6M frames.

| Name | Contains | Total Frames | Frames used |
|---|---|---|---|
| Ego4D | Ego4D | 418,578,043 | 2,790,520 |
| **Ego4D+M** (Manipulation) | Ego4D | 418,578,043 | 2,790,520 |
| | 100DOH | 99,899 | 99,899 |
| | SS-v2 | 25,209,271 | 315,115 |
| | Epic Kitchens | 19,965,439 | 332,757 |
| | | Total | 3,538,291 |
| **Ego4D+N** (Navigation) | Ego4D | 418,578,043 | 2,790,520 |
| | OpenHouse24 | 27,806,971 | 499,442 |
| | RealEstate10K | 10,000,000 | 303,087 |
| | | Total | 3,289,962 |
| **Ego4D+MN** (Manipulation, Navigation) | Ego4D+M | 3,538,291 | 3,538,291 |
| | OpenHouse24 | 27,806,971 | 499,442 |
| | RealEstate10K | 10,000,000 | 303,087 |
| | | Total | 4,340,820 |
| **Ego4D+MNI** (Manipulation, Navigation, ImageNet) | Ego4D+MN | 4,340,820 | 4,340,820 |
| | ImageNet | 1,281,167 | 1,281,167 |
| | | Total | 5,621,987 |

Table 7: Experiment Details of Training PVRs.

| Dataset Name | Epochs | Frames used |
|---|---|---|
| **Ego4D+N** (VIT-B) | 289 | 3,538,291 |
| **Ego4D+N** (VIT-L) | 289 | 3,538,291 |
| **Ego4D+M** (VIT-B) | 414 | 3,289,962 |
| **Ego4D+M** (VIT-L)) | 414 | 3,289,962 |
| **Ego4D+MN** (VIT-B) | 236 | 4,340,820 |
| **Ego4D+MN** (VIT-L) | 236 | 4,340,820 |
| **Ego4D+MNI** (VIT-B) | 182 | 5,621,987 |
| **VC-1** (**Ego4D+MNI** (VIT-L)) | 182 | 5,621,987 |

in their rank distributions. Notably, the **Ego4D+MNI** model consistently performs well across all benchmarks, and ranks among the top models.

## A.8 Additional Analysis of All Models Evaluated on CORTEXBENCH

Figure 6 (right) provides a complete picture of the rank distribution for all of the models evaluated in this study. Similarly, Table 8 provides results for all models evaluated in this study is collected.

## A.9 Additional Analysis of Scaling Model Size

Figure 7 illustrates that scaling model size has a positive effect on every benchmark and on fifteen out of the seventeen tasks.

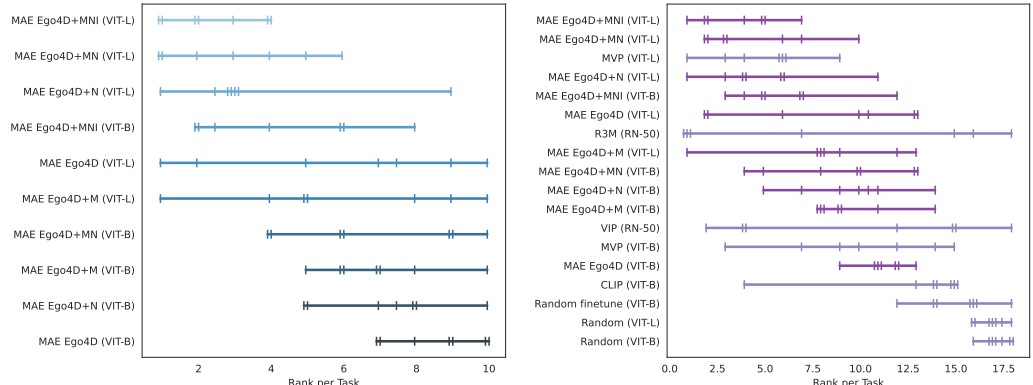

Figure 6: (left) Rank distribution per model - scaling hypothesis. (right) Rank distribution per model - existing PVRs and scaling hypothesis models

Table 8: The success rate for each task and each model we evaluate during the study before being aggregated by benchmark.

| | TASK | | | | | | | | | | | | | | | | |
| model | assembly | bin_picking | button_press | cheetah_run | drawer_open | finger_spin | hammer | imagenav | mobile_pick | move_cube | objectnav | pen | reach_cube | reacher | relocate | walker_stand | walker_walk |
|---|---|---|---|---|---|---|---|---|---|---|---|---|---|---|---|---|---|
| CLIP (VIT-B) | 70.7 | 68.0 | 48.0 | 22.7 | 100.0 | 74.6 | 90.7 | 52.2 | 49.8 | 40.1 | 56.6 | 72.0 | 83.8 | 89.9 | 22.7 | 64.9 | 25.4 |
| MAE Ego4D (VIT-B) | 81.3 | 76.0 | 80.0 | 29.1 | 100.0 | 76.9 | 93.3 | 64.0 | 57.4 | 54.0 | 46.8 | 74.7 | 82.6 | 79.8 | 22.7 | 84.3 | 50.5 |
| MAE Ego4D (VIT-L) | 98.0 | 84.0 | 84.0 | 20.7 | 100.0 | 76.5 | 98.7 | 55.8 | 67.6 | 57.0 | 47.6 | 76.0 | 82.4 | 71.9 | 24.0 | 78.4 | 56.3 |
| MAE Ego4D+M (VIT-B) | 76.0 | 58.7 | 84.0 | 31.9 | 100.0 | 75.5 | 98.7 | 65.8 | 59.8 | 57.5 | 47.4 | 77.3 | 80.7 | 89.3 | 25.3 | 80.7 | 44.3 |
| MAE Ego4D+M (VIT-L) | 89.3 | 73.3 | 84.0 | 33.5 | 100.0 | 75.6 | 94.7 | 65.5 | 68.6 | 47.2 | 47.3 | 74.7 | 82.1 | 85.8 | 29.3 | 76.2 | 52.3 |
| MAE Ego4D+MN (VIT-B) | 82.7 | 74.7 | 77.3 | 32.0 | 100.0 | 77.5 | 92.0 | 68.9 | 58.6 | 62.1 | 52.8 | 73.3 | 78.5 | 85.6 | 24.0 | 84.1 | 41.8 |
| MAE Ego4D+MN (VIT-L) | 93.3 | 70.7 | 74.7 | 38.1 | 100.0 | 77.0 | 94.7 | 69.1 | 61.2 | 62.4 | 58.4 | 78.7 | 82.4 | 91.7 | 26.7 | 83.0 | 58.9 |
| MAE Ego4D+MNI (VIT-B) | 88.0 | 78.7 | 82.7 | 32.3 | 100.0 | 76.0 | 98.7 | 67.9 | 60.6 | 60.6 | 55.4 | 76.0 | 83.9 | 82.6 | 32.0 | 83.5 | 44.7 |
| MAE Ego4D+MNI (VIT-L) | 88.0 | 84.0 | 80.0 | 32.8 | 100.0 | 76.8 | 92.0 | 70.3 | 63.2 | 60.2 | 60.3 | 80.0 | 83.3 | 88.0 | 38.7 | 83.3 | 53.7 |
| MAE Ego4D+N (VIT-B) | 86.7 | 76.0 | 73.3 | 28.1 | 100.0 | 75.8 | 96.0 | 68.7 | 59.4 | 54.1 | 54.7 | 77.3 | 81.6 | 78.7 | 22.7 | 72.4 | 42.6 |
| MAE Ego4D+N (VIT-L) | 89.3 | 73.3 | 89.3 | 33.3 | 100.0 | 76.2 | 93.3 | 70.5 | 65.2 | 52.7 | 57.4 | 76.0 | 81.1 | 88.6 | 32.0 | 83.4 | 50.7 |
| MVP (VIT-B) | 92.0 | 73.3 | 92.0 | 33.9 | 100.0 | 76.9 | 98.7 | 64.7 | 56.0 | 44.3 | 51.2 | 69.3 | 75.0 | 86.3 | 26.7 | 84.7 | 47.9 |
| MVP (VIT-L) | 89.3 | 78.7 | 70.7 | 36.9 | 100.0 | 76.4 | 98.7 | 68.1 | 65.4 | 63.4 | 55.0 | 76.0 | 84.8 | 90.2 | 30.7 | 83.2 | 59.3 |
| R3M (RN-50) | 97.3 | 93.3 | 89.3 | 66.1 | 100.0 | 77.1 | 100.0 | 30.6 | 33.2 | 51.9 | 22.6 | 81.3 | 86.5 | 98.4 | 65.3 | 93.8 | 70.1 |
| Random (VIT-B) | 0.0 | 0.0 | 2.7 | 0.4 | 0.0 | 0.1 | 0.0 | 42.1 | 10.8 | 41.3 | 19.2 | 4.0 | 74.3 | 23.4 | 0.0 | 22.7 | 4.0 |
| Random (VIT-L) | 0.0 | 0.0 | 0.0 | 0.5 | 0.0 | 0.2 | 2.7 | 45.2 | 20.6 | 39.4 | 19.3 | 5.3 | 74.9 | 19.9 | 0.0 | 20.1 | 4.6 |
| Random finetune (VIT-B) | 61.3 | 34.7 | 20.0 | 10.2 | 40.0 | 48.6 | 93.3 | 62.5 | 47.6 | 37.6 | 28.5 | 73.3 | 74.5 | 26.8 | 14.7 | 73.6 | 58.1 |
| VIP (RN-50) | 93.3 | 76.0 | 88.0 | 53.2 | 100.0 | 76.1 | 93.3 | 48.8 | 7.2 | 47.2 | 26.4 | 81.3 | 86.2 | 83.2 | 26.7 | 86.6 | 63.4 |

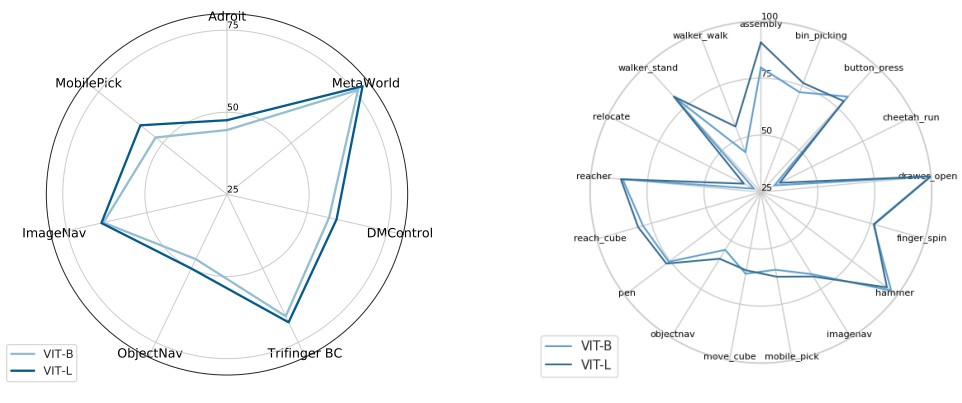

(a) scaling model size per benchmark

(b) scaling model size per task

Figure 7: Scaling model size has a positive effect on (a) every benchmark and on (b) fifteen out of the seventeen tasks.

## A.10 Attention Visualizations of VC-1

To visualize the attention we apply a mean pooling operation to the attention matrices of the ViT encoder's final layer during inference for downstream tasks. The resulting values are then overlaid onto the image.

We start by noticing the effect of MAE pre-training; frozen VC-1 attention maps appear to focus on the contours and general features of the image. We hypothesize that this results from the MAE

reconstruction-based training objective, as contours provide essential information for reconstructing images.

Additionally, we study the attention maps after end-to-end fine-tuning of VC-1 on the downstream tasks. The attention appears to focus on regions of the image that are important for the task (e.g., the objects being manipulated). Thus, through adaptation (via E2E fine-tuning), the model learns to drop attention on areas irrelevant to the specific task.

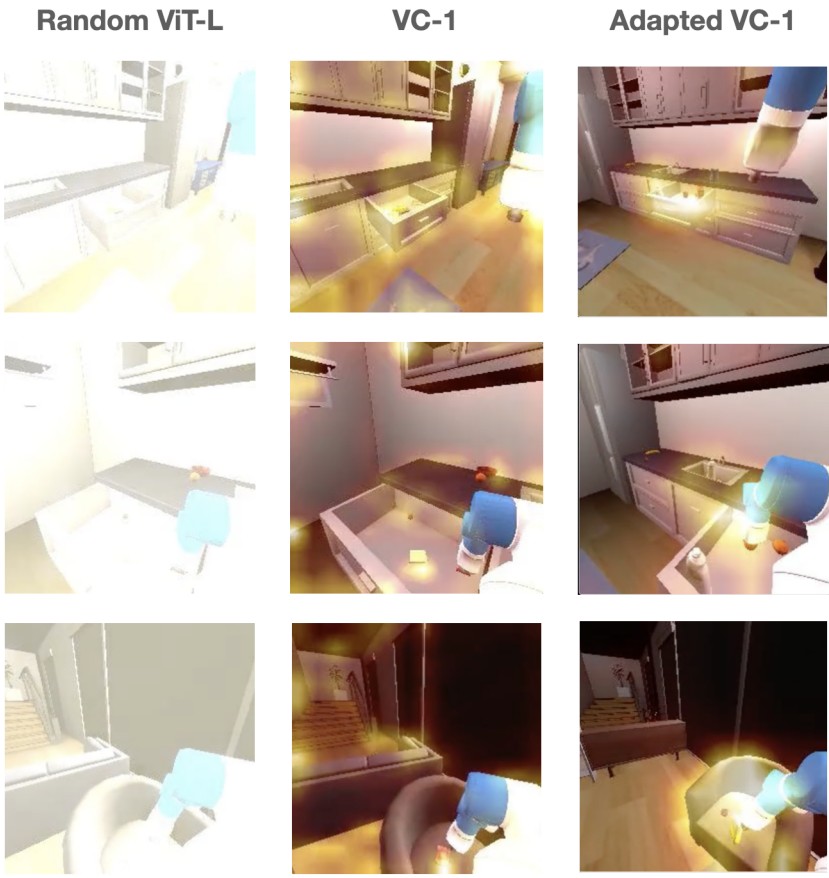

Figure 8: Attention Visualization: (left) Random ViT-L; (middle) VC-1 frozen; (right) VC-1 E2E finetuned. We overlay the mean attention matrix in the last layer of the ViT encoder in one of our tasks -MobilePick-. We notice the effect of MAE pre-training on VC-1: The attention focuses in general features of the image; and of task-adaptation: the attention concentrates in task-specific regions of the image

## A.11 TriFinger Hardware Experiment Setup

We carried out experiments on the real TriFinger robot (shown in Figure 9) for the `Push-Cube` task, after training a model using behavior cloning on 30 real-world demonstrations. The success for `Push-Cube` is defined as the distance which the cube is able to travel relative to its starting distance to the goal, similar to [43]. The episode length for this task is 20 steps. For each model, we chose 1 seed and ran it on 12 different start and goal configurations, mostly centered in the arena.

The action space for this environment consists of end-effector displacements for the three fingers. The motors are controlled at a frequency of 1kHz and the action sent to the robot is a 9 dimensional vector specifying the joint torques. We use an Intel RealSense camera, positioned next to the table overlooking the scene, similar to how our simulation image observations are captured. To detect the cube, we make sure that the green side is always facing upwards and detect the center of that face, and use that to track the cube's position in the bowl.

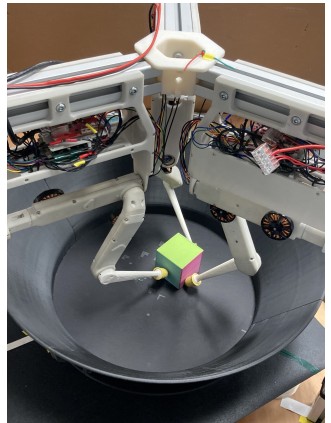

Figure 9: TriFinger Push Cube task

## A.12 Franka Hardware Experiment Setup

We conduct behavior cloning experiments on a manipulation platform comprising a Franka arm and a Robotiq gripper fitted with Festo Adaptive Gripper Finger. Figure 10 shows the task setups:

- The reaching task requires controlling the end-effector to reach an randomly selected point between $(-0.3, 0.4, 0.2)$ and $(0.3, 0.4, 0.2)$ in the robot frame within 5cm error given a goal image prompt.

- The bottle pickup task requires reaching and picking up a bottle with the gripper. The bottle is randomly positioned on a 10 cm line, same orientation).

- The close-drawer and toaster plunge tasks are similar to the Cacti setup [72].

For all the tasks, the observations are single RGB camera image (420x224 resolution) and proprioceptive signals (7 degree-of-freedom joint angles and 1 degree-of-freedom gripper width). A 3-hidden layer, each with 256 hidden nodes, multi-layer perceptron policy takes PVR encoded images concatenated with scaled proprioceptive signals as input and predicts the desired absolute joint angles and gripper width as actions. A low-level 1kHz real-time joint-space PD controller follows the policy-generated desired trajectory. We specifically keep the gains low (half of the factory setting) so that the robot will not break anything even if it were commanded to hit the table. This results in poor trajectory tracking performance, yet both human teleoperators and learned policy can control the robot to complete the tasks. We use Robohive [73] as the middleware to interface with all sensors and robots.

Demonstrations are collected from human teleoperation using a Quest 2 controller. For PVR (frozen encoders), we use Adam optimizer with a learning rate $10^{-3}$ to train the policies. For fine-tuning, we use the same learning rate for policies but a lower learning rate ($10^{-5}$) for the visual encoders. Table 9 shows the behavior cloning success rates.

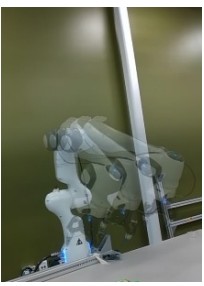 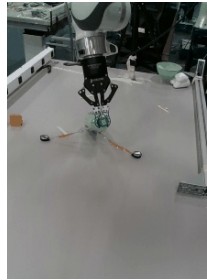 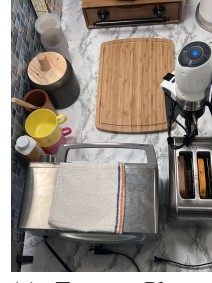 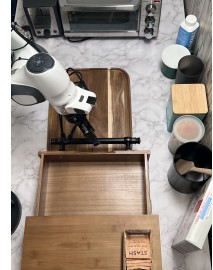

(a) Reaching Random Point    (b) Bottle Pickup    (c) Toaster Plunge task on a kitchen table-top setup    (d) Open Drawer task on a kitchen table-top setup

Figure 10: Franka manipulation tasks

Table 9: Franka manipulation task success rates.

|  | Reaching | Bottle Pickup | Open Drawer | Plunge Toaster | Success Count | Success % |
|---|---|---|---|---|---|---|
| Demos | 30 | 50 | 250 | 250 |  |  |
| Evals | 10 | 10 | 10 | 10 |  |  |
| VC-1 frozen | 80 | 100 | 60 | 40 | 28 | 70.0 |
| VC-1 E2E fine-tuning | 50 | 90 | 80 | 50 | 27 | 67.5 |
| VC-1 MAE adaption | 90 | 100 | 80 | 70 | 34 | 85.0 |
| MAE baseline | 30 | 10 | 50 | 50 | 14 | 35.0 |
| MVP | 70 | 100 | 30 | 20 | 22 | 55.0 |

