# OpenReview forum: "Where are we in the search for an Artificial Visual Cortex for Embodied Intelligence?"
_NeurIPS.cc/2023/Conference — NeurIPS 2023 poster_

### Official Review · Reviewer_on5Q · 2023-07-04

**Soundness:** 3 good
**Presentation:** 4 excellent
**Contribution:** 4 excellent
**Rating:** 8
**Confidence:** 4

**Summary:**

The paper introduces a study of pre-trained visual encoders for Embodied AI, and produces, among others, a new pre-trained ViT-L encoder (VC-1) using Masked Auto-Encoding on a dataset combining an expanded collection of egocentric video and ImageNet. VC-1, when adapted for agents solving 17 task types covering dexterous manipulation, locomotion, mobile manipulation, and navigation, performs best in average. The curated set of 17 task types is presented as a new benchmark named CortexBench. The results show that no visual encoder is universally dominant, and that just scaling the pre-training dataset size and diversity does not imply uniform improvements across benchmarks, refuting previous extrapolated results. Further experiments on real hardware are also included, showing the improvement in comparison with the best performing existing Masked Visual Pre-training visual encoder.

**Strengths:**

- Thorough study of pre-trained visual encoders for Embodied AI tasks across different pre-training schemes.
- Provides a reference model with high performance (especially after adaptation) across all considered task types.
- Thorough experimental section and introduction of the very relevant CortexBench benchmark.
- The paper reads really well.

**Weaknesses:**

- In an ideal world, it would have been wonderful to have found a single pre-trained model that could be used without adaptation with high performance across all tasks, but the actual outcome is thoroughly justified in the paper. One open question is whether, with larger and more diverse datasets, a single pre-trained model could eventually be the absolute winner before adaptation, but I agree this is out of scope for this paper.

**Questions:**

- Line 370 states "MAE adaptation does not help TriFinger performance, this is likely due to the mere 300 real robot images available for adaptation.". However, fine-tuning seems to do a good job in this case, contradicting the general consensus derived in Section 6. Is there an intuitive explanation?
- Why is prompting/conditioning VC-1 not considered as an adaptation method in the study? It might be a viable not-so-costly alternative to fine-tuning.
- Is the advantage of R3M indicating that known architectures might be far from optimal for the goal of having a single visual encoder able to feed all EAI tasks or is the scale of the pre-training datasets considered in the study the most likely reason?

**Limitations:**

The paper discusses adequately the limitations.

---

> ### Author Rebuttal · Authors · 2023-08-10
>
> Thank you for taking time to review our paper and sharing your thoughts. Please find responses to your questions/suggestions below. Please let us know if any additional clarifications are needed.
>
> > Line 370 states "MAE adaptation does not help TriFinger performance, this is likely due to the mere 300 real robot images available for adaptation.". However, fine-tuning seems to do a good job in this case, contradicting the general consensus derived in Section 6. Is there an intuitive explanation?
>
> There are broadly two reasons why a frozen visual encoder may improve via fine-tuning on task data: 1) closing the domain gap, 2) moving from task-independent features to task-specific features. MAE adaptation does the former, BC fine-tuning does the latter. Which among the two is more important could be task-dependent. Our results suggest that the task-specific fine-tuning was more important than closing any domain gap for TriFinger. We will add this discussion to the paper.
>
> > Why is prompting/conditioning VC-1 not considered as an adaptation method in the study? It might be a viable not-so-costly alternative to fine-tuning.
>
> It is unclear to us how to produce prompts for the tasks in CortexBench, as there is no standard approach that we are aware of. However, we’d be happy to add more comments if the reviewer could provide more details about the prompts they had in mind.
>
> > Is the advantage of R3M indicating that known architectures might be far from optimal for the goal of having a single visual encoder able to feed all EAI tasks or is the scale of the pre-training datasets considered in the study the most likely reason?
>
> We're not sure we fully understand the question. We believe you're asking what are important ingredients to training a general purpose visual encoder for EAI (architecture, pre-training objective, dataset diversity, dataset size, etc.). The short answer is that we believe all of these matter, and finding the best combination is an open research problem. Our work shows that pre-training dataset diversity definitely matters, and that (thus far) a ViT backbone slightly outperforms other backbones (on average).

---

> > ### Comment · Reviewer_on5Q · 2023-08-14
> > **Thank you for the rebuttal**
> >
> > I would like to thank the authors for the detailed responses to the reviewers' questions. I reopen some questions that I think were misunderstood or unclear in my original review.
> >
> > - The first unclear question was regarding fine-tuning alternatives:
> >
> > > It is unclear to us how to produce prompts for the tasks in CortexBench, as there is no standard approach that we are aware of. However, we’d be happy to add more comments if the reviewer could provide more details about the prompts they had in mind.
> >
> > I forgot to add the corresponding lines in my original review, but I actually referred to Lines 297-298 in the original manuscript:
> >
> > > We use a broad definition of adaptation [50], which, in the context of large pre-trained foundation models, can take several forms from simple prompting [51] ...
> >
> > Apologies if I am mistaken, but this line of research is no further discussed (or explicitly discarded) for the rest of the study, so it might have been worth it to add on line explaining why this mentioned option is not (or cannot be) considered, despite its potential advantages/interest in other domains.
> >
> > - The second unclear question was about the potential advantages of R3M compared to VC-1 and possible impact in upcoming models. The original manuscript (Lines 288-291) states:
> >
> > > When compared to R3M, VC-1 demonstrates superior performance on average and on 4 of 7 benchmarks (Figure 4). R3M outperforms VC-1 on Adroit, MetaWorld and DMControl benchmarks. It is unclear whether this gap is caused by the different training objective, pre-training dataset, or backbone.
> >
> > In recent years, transformers have been thoroughly used as the standard architecture in a large (and extremely varied) amount of domains, principally due to their advantageous scalability properties. However, it was impressive to see R3M (whose architecture is based on ResNet) perform so well in comparison to VC-1 based on transformers. Without being very concrete in my question (and obviating other important factors like the ones correctly mentioned by the authors in their response), I was meaning to start a discussion on the importance of research of new architectures that could hopefully bring the best of both worlds (inductive biases and scalability) in a principled way. In any case, I am happy with the provided response

---

> > > ### Author Response · Authors · 2023-08-20
> > > **Thank you for the clarifications**
> > >
> > > Thank you for the clarifications. These are great points and we address both below.
> > >
> > > > it might have been worth it to add on line explaining why this mentioned option [prompting] is not (or cannot be) considered
> > >
> > > Agreed. The PVRs studied in this work were not originally designed for few-shot prompting. Thus, adding prompting to these methods remains an open research question. We will add a note to the paper.
> > >
> > > > I was meaning to start a discussion on the importance of research of new architectures that could hopefully bring the best of both worlds (inductive biases and scalability) in a principled way. In any case, I am happy with the provided response
> > >
> > > We agree! We believe research on new architectures is a fruitful direction to pursue, and is well motivated by the results of R3M and VC1 on CortexBench. We will discuss this interesting observation in the paper.

---

### Official Review · Reviewer_5TRU · 2023-07-05

**Soundness:** 3 good
**Presentation:** 4 excellent
**Contribution:** 4 excellent
**Rating:** 7
**Confidence:** 5

**Summary:**

•	This paper presents the largest and most comprehensive empirical study of pre-trained visual representations for Embodied AI based on proposed CORTEXBENCH.

•	This paper focuses on studying the effect of pre-training data size and diversity for PVRs, further proposing VC-1.

•	This paper shows that task- or domain-specific adaptation of VC-1 leads to substantial gain by E2E fine-tuning or MAE adaptation.


**Strengths:**

•	This paper proposes a comprehensive benchmark CORTEXBENCH for pre-trained visual representation evaluation of embodied intelligence, which is very valuable and meaningful.

•	This paper studies the effect of pre-training data size and diversity for PVRs, which is novel and insightful.


**Weaknesses:**

•	There may be some typos in experiments, such as ‘Trifinger by+7.4 (72.7 → 80.1)’ in the 343 line, and according to the table, it should be Trifinger (71.7 → 80.6). Please carefully check.

•	Although I think this work has been done well enough. However, there are still many questions that can be explored: based on ViT and the same data, which is better for MIM and time-contrastive video-language alignment. Is ResNet better than ViT when few-shot imitation learning?

•	This paper only attempts to adapt VC-1, but there are no corresponding experiments for R3M and CLIP.

•	I strongly recommend including a concise introduction to multimodal-based pre-training visual representation in the related works section, citing relevant sources such as [1-5]. This is crucial as multimodal presentation learning now plays a pivotal role in numerous vision tasks.
[1] Instruction-Following Agents with Multimodal Transformer. [2] Language-Driven Representation Learning for Robotics. [3] Open-World Object Manipulation using Pre-trained Vision-Language Models. [4] Pave the Way to Grasp Anything: Transferring Foundation Models for Universal Pick-Place Robots. [5] LIV: Language-Image Representations and Rewards for Robotic Control. Among them, [3] and [4] explicitly adapt the middle-level output of the pre-trained model through larger policy models, which also utilize the raw image as input.


**Questions:**

•	I'm curious how you choose specific tasks under each benchmark. Does this follow some basis?

•	I'm curious why you did not model the temporal information in policy model. For example, adding temporal attention in ViT, even if utilizing a large amount of video data for pre-training.


**Limitations:**

•	Although the evaluation of this work is already very comprehensive, I believe that 17 tasks are not sufficient. I found that even for the same benchmark, different tasks exhibit significant differences.

---

> ### Author Rebuttal · Authors · 2023-08-10
>
> Thank you for taking time to review our paper and sharing your thoughts. Please find responses to your questions/suggestions below. Please let us know if any additional clarifications are needed.
>
> > Add a concise introduction to multimodal-based pre-training visual representation in the related works section, citing relevant sources such as [1-5].
>
> Thanks! We would be happy to add a discussion of these works to the paper.
>
> > Although I think this work has been done well enough. However, there are still many questions that can be explored: based on ViT and the same data, which is better for MIM and time-contrastive video-language alignment. Is ResNet better than ViT when few-shot imitation learning?
>
> These are good questions that we are also interested in. However, given the scale of experiments required to answer the research questions posed in sections 4, 5, and 6, these additional questions fell beyond the scope of this work. We speculate that using temporal information for learning visual representations (as done in time-contrastive learning) will be useful, and might be a good direction to pursue in future work.
>
> > Although the evaluation of this work is already very comprehensive, I believe that 17 tasks are not sufficient.
>
> The tasks included in CortexBench were chosen to cover a wide range of embodied AI (EAI) applications, and are of substantial interest to the EAI community as evidenced by the body of work that uses and reports on these tasks. That said, we agree that additional tasks could be added in future studies. As discussed in L633-L636 in the Appendix, "in proposing a benchmark, we sought to find a balance between task diversity and the computational resources required for evaluation. However, new and challenging benchmarks in embodied AI, such as those presented in [55], continue to emerge and may merit inclusion in future studies to track progress in this field.”
>
> > I'm curious how you choose specific tasks under each benchmark. Does this follow some basis?
>
> Yes, we selected tasks following the precedent and trends seen in recent works that have studied each of the individual benchmarks. For example, the MetaWorld tasks were selected “following the evaluations in [7]” (L157), the DMC tasks “following [6]” (L160), and in Adroit “we study the two hardest tasks” (L149-150).
>
> > I'm curious why you did not model the temporal information in policy model.
>
> Most of the downstream policies do model temporal information either through frame stacking (Adroit, MetaWorld, DMC) or by using a recurrent neural network (ObjectNav, ImageNav, Mobile Pick). We will make this more clear in the paper.
>
> > Typos in experiments
>
> Thanks for pointing this out. We will correct these typos.

---

### Official Review · Reviewer_aWtL · 2023-07-06

**Soundness:** 3 good
**Presentation:** 3 good
**Contribution:** 3 good
**Rating:** 5
**Confidence:** 5

**Summary:**

This paper focuses on the analysis of large-scale pre-trained vision models for interactive tasks. The authors curate CortexBench which includes tasks covering dexterous manipulation, navigation, and also scene-level interaction as the testing benchmark. Further experiments were made on testing existing pre-trained vision models on this task. The authors also control the dataset type, the scale used in pertaining and proposed their VC-1 model pre-trained in an MAE fashion which shows superior performance compared to baselines. Additional analysis were made showing the effectiveness of egocentric images for interaction and also gap between existing pre-trained vision models and the desired "visual cortex" representation.

**Strengths:**

This paper conducts solid experiments on testing existing pre-trained visual representations (PVR) on their own curated interaction benchmark CortexBench. Both the evaluation benchmark and released checkpoints (as promised by the authors) will be extremely beneficial for embodied AI research. The overall analysis is technically sound with proper conclusions made, pointing to future directions for PVR studies.

**Weaknesses:**

One concern of this paper is that except for the conclusion pre-training with egocentric frames is beneficial for interactive EAI tasks, I do not see signature conclusions that could be drawn from the current analysis. After reading this manuscript, I do not feel that the question in the paper title has been properly addressed:
- First, there is no task analysis for the curated tasks in CortexBench, for a proper visual cortex, to what extent should we expect with "visual cortex"? are these tasks equally important? are there any relationships between these tasks?

- Second, since the current VC-1 model still does not universally dominate either task, what is missing on the training/model side? is it missing for example value condition in VIP or some embodied experience?

- Third, as a pre-trained vision representation, what should we expect from VC-1 in addition to its capabilities in CortexBench? Should we expect the model to at least have some sort of objectness as DINO? Can this pre-trained model be used in more general vision settings?

**Questions:**

Please see the weakness section for questions.

**Limitations:**

The authors have properly addressed the limitations of this paper.

---

> ### Author Rebuttal · Authors · 2023-08-10
>
> Thank you for taking time to review our paper and sharing your thoughts. We have addressed your concerns below. Please let us know if any additional clarifications are needed.
>
> > First, there is no task analysis for the curated tasks in CortexBench.
>
>
> Because CortexBench is curated from existing benchmarks (Sec. 3.1) a detailed analysis of each task can be found in the respective papers. The tasks span a wide spectrum of EAI. They vary in visual complexity, task complexity, task domain, skills required, agent embodiments, and how policies are trained. In that sense we have a good coverage of various EAI problems.
>
> > Are these tasks equally important?
>
> In this work we do not take a position on which tasks are more or less “important” – that is for the community to decide. However, all of the tasks are widely studied in the community as evidence by the citation counts for each benchmark: Adroit [12] (801 citations), MetaWorld [11] (648 citations), DeepMind Control [10] (420 citations), TriFinger [13] (43 citations), Habitat [14] (931 citations), and Habitat 2.0 [15] (277 citations).
>
> > Are there any relationships between these tasks?
>
> The overarching relationship is that these tasks have been deemed important and interesting by the EAI community as evidenced by the body of work that has gone into studying each of the tasks separately. In this work, we use the collection to draw new conclusions about existing and new PVRs.
>
> At a lower level, there are relationships in the way goals are specified and the policy learning algorithms that have led to SoTA performance in prior work as illustrated in Table 1. Finally, there are commonalities in the behaviors required. For example, ObjectNav, ImageNav, and Mobile Pick all require navigation. We will add this to the discussion.
>
> > VC-1 does not universally dominate; what is missing? Is it missing for example value condition in VIP or some embodied experience?
>
> Great question! Value conditioning or embodied experience may help and would be quite interesting to explore.
>
> Additionally, the visual embedding in VC-1 is for a single frame, so most of the downstream policies offload the environmental representation construction to a generic RNN or frame stacking. There are several avenues to consider for improving representations including using videos/temporal axis, 3D spatial priors, objectness, etc. We will add these future directions to the discussion section.
>
> > What should we expect from VC-1 in addition to its capabilities in CortexBench? Should we expect the model to at least have some sort of objectness as DINO? Can this pre-trained model be used in more general vision settings?
>
> We expect that VC-1 will be useful in other EAI tasks because of its effectiveness on the diverse range of tasks in CortexBench. Additionally, we provide some preliminary analysis of the visual features in appendix A.10, where we find, for example, that after adaptation the visual attention focuses more on task-relevant parts of the image such as task-relevant objects. In general, we are excited for the community to use our open-sourced models on additional tasks and settings.

---

> > ### Comment · Reviewer_aWtL · 2023-08-21
> > **Post-rebuttal response**
> >
> > The authors' responses have clarified some of my concerns. I still feel that there are still more analyses to be made to make this benchmark design more meaningful and complete (instead of selecting the most common or popular existing EAI tasks). Therefore, I'm keeping my original rating as borderline accept, however, I will not argue for rejection.

---

### Official Review · Reviewer_nrzP · 2023-07-06

**Soundness:** 2 fair
**Presentation:** 2 fair
**Contribution:** 1 poor
**Rating:** 2
**Confidence:** 4

**Summary:**

This paper evaluates different pre-trained visual representations w.r.t. their capability to serve as foundation models for Embodied AI. To do so, they compile a dataset (CortexBench), with 17 simulated EAI tasks. The intuition here is that a single pre-trained visual representation which would perform well across all tasks could be considered as an Artificial Visual Cortex. The paper collects several egocentric video datasets starting with Ego4D,  and sequentially adding datasets to create different variations trained with different training datasets. Their proposed artificial visual cortex (VC-1) is trained on all datasets (ViT-L trained with Ego4D + MNI). They also propose adaptations to VC-1 to improve its capability to perform well across different datasets. Finally a proof of concept hardware experiment using VC-1 to learn policies with imitation learning for few shot manipulation tasks.

**Strengths:**

1. Exhaustive training with lots of variations with architectures and increasingly growing dataset.

2. Evaluation conducted on a large scale dataset with a diverse set of EAI tasks.

3. Writing: The paper is easy to read and follow. The figures do a good job of illustrating the dataset and the experiments. Also, the.

**Weaknesses:**

1. No data contribution: CortexBench is a compilation of existing datasets, and it's not clear what new it brings to the table beyond what's already out here. From the introduction, it initially seemed as thought it's newly proposed benchmark in this paper. However, it seems to be a set of already existing datasets. Why is it being claimed/renamed as a new dataset?

2. No new findings:

- The first negative result is that specialized pre-trained models are better - that is, in-distribution models perform better than out-of-distribution ones (lines 42-45). This is well known in the machine learning community, and thus, is not a new finding.
- Lines 70-85 can be summarized as saying: More data is better, but OOD data is a problem. This too is a well known property of deep learning vision models.
- Naively scaling size and diversity: The paper does not study data diversity explicitly, but rather only

3. Data diversity and Dataset size are not controlled: When the new datasets are added to the foundation models, are the number of images held constant? I didn't see this detail in the paper. Thus, it is unclear what helped - dataset size, or diversity? To control data diversity independently, a control would be to reduce number of images per dataset as # datasets is increased to isolate the effect of size and diversity.

4. The adaptive VC-1 does not really work well: of the datasets with prior results in Table 5, VC-1 does better for only 2 task classes. Without adaptation too, it could be trained to do well on one of these task classes alone. Thus, the adaptation took it to performing well from 1 task to 2 task classes. Furthermore, there are no error bars in these results which makes it hard to evaluate the reliability of these findings given the stochasticity of the models.

5. Claim too grandiose: Calling a pre-trained visual recognition model an artificial cortex is a very far fetched claim on several levels of abstraction. Firstly, the cortex does much more than EAI: so testing it only on a small set of functionalities of the cortex means there's a mismatch at a behavioral level between the cortex and the artificial cortex. Secondly, by not comparing with any neural data, there is a mismatch at the mechanistic level as well. A more objective representation of the work done here is that this paper investigates --- Can a single pre-trained representation perform well across a diverse set of EAI tasks?

**Questions:**

Please refer to weaknesses above for questions.

---

> ### Author Rebuttal · Authors · 2023-08-10
>
> Thank you for taking time to review our paper. We believe there may be some major misunderstandings about our submission, and we would like to take this opportunity to clarify.
>
> > No data contribution. From the introduction, it initially seemed as thought it's newly proposed benchmark in this paper. Why is it being claimed/renamed as a new dataset?
>
> We do not claim a data contribution. As stated in the intro, "we *curate* CortexBench" and we cite the sources in L34-35; section 3 describes details. A name is used to succinctly refer to the curation in the exposition and to the associated software.
>
> Our contributions are stated upfront in abstract L16-20: "Overall, this paper presents no new techniques but a rigorous systematic evaluation, a broad set of findings about PVRs (that in some cases, refute those made in narrow domains in prior work), and open-sourced code and models (that required over 10,000 GPU-hours to train) for the benefit of the research community."
>
> Furthermore, we agree with reviewer aWtL: "Both the evaluation benchmark and released checkpoints (as promised by the authors) will be extremely beneficial for embodied AI research."
>
> > specialized pre-trained models are better - that is, in-distribution models perform better than out-of-distribution ones (lines 42-45).
>
> This interpretation is not inline with our findings. None of the PVRs considered in this work (except in Section 6) were trained on images from the evaluation tasks or simulators, and thus are all out-of-domain by definition. What we're seeing is that previous work had specialized on evaluating PVRs on a small set of tasks and shown strong results, but those results do not carry over to a wide distribution of tasks, which is a new finding not known in prior work. There is a broader question here of “what exactly is a domain”, but that is beyond the scope of this work (we use standard definitions).
>
> > Lines 70-85 can be summarized as saying: More data is better, but OOD data is a problem.
>
> Not quite. As pointed out above, all training data we use is OOD, so what this finding is saying is that it is difficult to know a priori which data should be included or excluded from the pre-training set; and without our experiments, we would not know about the existence of this problem. As we say in L84-85: "[our] broad evaluation refutes a naive extrapolation of the positive scaling trends observed in prior work on robot manipulation [8]."
>
> > Data diversity and Dataset size are not controlled: When the new datasets are added to the foundation models, are the number of images held constant? I didn't see this detail in the paper.
>
> Yes, the training dataset size is held constant to support claims about data diversity. The number of images per dataset is provided in Table 3 on page 5 and a more detailed breakdown is provided in Table 6 in the Appendix.
>
> Specifically, comparisons between the **Ego4D+M** and **Ego4D+N** training datasets – both containing 3.5M frames – support the dataset diversity claims. As stated on L288 “[Ego4D+N] is similar in size to Ego4D+M (3.5M frames) but is more diverse because it contains a larger proportion of navigation data than the manipulation-centric datasets Ego4D and Ego4D+M.”
>
> > of the datasets with prior results in Table 5, VC-1 [adapted] does better for only 2 task classes.
>
> It is not clear why nearly half the experiments in Table 5 are being ignored in making this claim. To recap, Table 5 presents results on 7 simulation benchmarks ("task classes") and 2 hardware tasks. Prior published results exist for 5 out of 9 benchmarks because as we describe in the abstract, intro, and experiments, prior work has focused on narrow task domains; our contribution is to systematically benchmark existing PVRs on this broad set of tasks (which is what Table 2 and row 2 in Table 5 provide).
>
> When all tasks are considered, VC-1 adapted (via MAE adaptation in low-shot IL or via fine-tuning in large-scale RL) outperforms VC-1 (without adaptation) on all 7 benchmarks. Additionally, as stated on L351-354, “Furthermore, on MetaWorld, DMControl, and TriFinger, VC-1 with MAE adaptation (row 6 [in Table 5]) is comparable with the best known results (SoTA) and the best results from previous sections (rows 1 and 2 [in Table 5]). Similarly, on ImageNav and Mobile Pick, VC-1 with E2E fine-tuning (row 5 [in Table 5]) matches or exceeds the best results.”
>
> > No error bars in Table 5.
>
> Thank you for this suggestion. We have added error bars to Table 5. We find that the error bars for the adapted versions of VC-1 (Table 5 rows 5 and 6) are very similar to the error bars we report for VC-1 in Table 4 row 11. Accordingly, all of the conclusions drawn in the paper remain unchanged.

---

### Author Rebuttal · Authors · 2023-08-10

We thank the reviewers for their thoughtful comments. We want to begin with a few high-level observations.

There appears to be consensus that our work presents “a large scale dataset with a diverse set of EAI tasks” (nrzP) – i.e., CortexBench. Several reviewers believe CortexBench will be “extremely beneficial for embodied AI research” (aWtL), is “valuable and meaningful” (5TRU), and is “very relevant” (on5Q). Similarly, there appears to be agreement that we perform a “thorough study” (on5Q) with “exhaustive training with lots of variations” (nrzP).

However, one reviewer disagrees with the others on the novelty and the degree of surprise in the findings from our study.

On one hand, several reviewers find that we present experiments that are “novel and insightful” (5TRU) and “technically sound with proper conclusions made, pointing to future directions for PVR studies.” (aWtL). On the other hand, one reviewer expresses that the findings are “well known in the machine learning community” (nrZP). The disagreement here lies in the perceived newness of our findings rather than the technical substance of our research.

We contend that these are precisely the kinds of papers that should be presented at a conference like NeurIPS because they provoke thought, stimulate discussion, and provide rigorous benchmarking that forms the foundation for future work. In our case, we present a rigorous empirical study that pushes the boundaries of understanding in the field of Embodied AI.

If one of the reviewers is skeptical about certain aspects of our work, that's part of the scientific discourse—we can let the broader community and future research be the judge of the long-term impact.

Below, we address specific concerns raised by each reviewer to provide further clarity and address any misunderstandings.

---

### Decision · Program_Chairs · 2023-09-21

**Decision:**

Accept (poster)

**Comment:**

This work received 2 positive, 1 borderline positive and 1 strongly negative review.

The positive reviewers appreciated
- exhaustive evaluation of pretrained visual encoders across a variety of Embodied AI tasks
- study of the effect of diversity and scale
- release of checkpoints (as promised by authors)

However, one reviewer (nrzP) expressed strong concerns about:
1. whether CortexBench may be considered a contribution since it is a combination of existing benchmarks?
2. are there really any new findings in this work?
3. while studying the effect of dataset size and diversity, the two factors are entangled. For instance, to study the effect of diversity, the reviewer suggests keeping the total number of images the same while varying the proportion of different tasks in the mixture. The diversity question really is - *Given a fixed budget of training images, is it better to have fewer images per task but a lot of tasks or a huge number of images per task from fewer tasks*?
4. Grandiose claims - is the pretrained model in this work any more a visual cortex than any other self-supervised pretrained model?

The AC finds reviewer nrzP’s concerns reasonable and commends them for their dedication towards a thorough review. Having read all reviews and the rebuttal, here’s the AC’s take on each of these 4 issues:
1. Even if CortexBench is constituted entirely of existing benchmarks, unifying several benchmarks and making it easier for other researchers to evaluate their models and representations could be considered a contribution. However, the AC believes the main merit of this work is not in the benchmark itself but in running a *large scale empirical study* focused on the question - *could scaling MAE across diverse benchmarks lead to good universal visual representations for EAI tasks*?
2. The AC somewhat agrees that the work doesn’t present a completely unexpected or novel finding and some of the results are indeed mixed. However, the AC believes the results shown in this paper could serve as a useful reference point for future studies.
3. The reviewer does have a valid point here, and the AC believes the authors didn’t adequately address this concern.
4. The AC somewhat agrees with this sentiment, but while the title and introduction might appear a bit grandiose, the technical portions of the paper seem well-grounded. Some readers might enjoy an imaginative introduction while others don’t - a matter of taste.

Overall, weighing the pros and the cons, the AC believes this work is **worth accepting** as a useful reference point for future research on building universal visual representations for generalist Embodied agents. Besides the empirical study as presented in the paper, the evaluation benchmark and checkpoints could also be of value.

The AC would encourage the authors to incorporate reviewers’ suggestions and discussion to increase the impact of their work.